# Online Policy Optimization for Robust Markov Decision Process

**Jing Dong** [*1]      **Jingwei Li** [*2]      **Baoxiang Wang** [*1,3]      **Jingzhao Zhang** [*2,4]

[1]The Chinese University of Hong Kong, Shenzhen
[2]Institute for Interdisciplinary Information Sciences, Tsinghua University
[3]Vector Institute
[4]Shanghai Qizhi Institute

## Abstract

Reinforcement learning (RL) has exceeded human performance in many synthetic settings such as video games and Go. However, real-world deployment of end-to-end RL models is less common, as RL models can be very sensitive to perturbations in the environment. The robust Markov decision process (MDP) framework—in which the transition probabilities belong to an uncertainty set around a nominal model—provides one way to develop robust models. While previous analysis for robust MDP shows RL algorithms are effective assuming access to a generative model, it remains unclear whether RL can be efficient under a more realistic online setting, which requires a careful balance between exploration and exploitation. In this work, we consider online robust MDP by interacting with an unknown nominal system. We propose a robust optimistic policy optimization algorithm that is provably efficient. To address the additional uncertainty caused by an adversarial environment, our model features a new optimistic update rule derived via Fenchel conjugates. Our analysis establishes the first regret bound for online robust MDPs.

## 1 INTRODUCTION

The rapid progress of reinforcement learning (RL) algorithms enables trained agents to navigate around complicated environments and solve complex tasks. The standard reinforcement learning methods, however, may fail catastrophically in another environment, even if the two environments only differ slightly in dynamics Farebrother et al. [2018], Packer et al. [2018], Cobbe et al. [2019], Song et al.

[2019], Raileanu and Fergus [2021]. In practical applications, such mismatch of environment dynamics are common and can be caused by a number of reasons, e.g., model deviation due to incomplete data, unexpected perturbation and possible adversarial attacks. Part of the sensitivity of standard RL algorithms stems from the formulation of the underlying Markov decision process (MDP). In a sequence of interactions, MDP assumes the dynamic to be unchanged, and the trained agent to be tested on the same dynamic thereafter.

To model the potential mismatch between system dynamics, the framework of robust MDP is introduced to account for the uncertainty of the parameters of the MDP Satia and Lave Jr [1973], White III and Eldeib [1994], Nilim and El Ghaoui [2005], Iyengar [2005]. Under this framework, the dynamic of an MDP is no longer fixed but can come from some uncertainty set, such as the rectangular uncertainty set (e.g. in Iyengar [2005], Nilim and El Ghaoui [2005]), centered around a nominal transition kernel. The agent can interact with a nominal transition kernel to learn a policy, which is then evaluated on the worst possible transition from the uncertainty set. Therefore, instead of searching for a policy that may only perform well on the nominal transition kernel, the objective is to find the worst-case best-performing policy.

Robust MDP can be viewed as a dynamical zero-sum game, where the RL agent tries to choose the best policy while nature imposes the worst possible dynamics. When the environment transition and reward is known, solving the robust MDP problem is tractable under suitable assumptions due to the aforementioned works. When the information about the environment is missing, if a generative model (also known as a simulator) of the environment or a suitable offline dataset is available, one could obtain a $\epsilon$-optimal robust policy with $\tilde{O}(\epsilon^{-2})$ samples Qi and Liao [2020], Panaganti and Kalathil [2022], Wang and Zou [2022], Ma et al. [2022].

However, the presence of a generative model is rare in real applications. Therefore, in this work, we consider an *online*

---

[*]Authors are ordered alphabetically. Corresponding to: Jing Dong, Jingwei Li, Baoxiang Wang, Jingzhao Zhang.

*Accepted for the 40[th] Conference on Uncertainty in Artificial Intelligence* (UAI 2024).

Table 1: Comparisons of previous results and our results, where $S$, $A$ are the size of the state space and action space, $H$ is the length of the horizon, $K$ is the number of episodes, $\rho$ is the radius of the uncertainty set and $\epsilon$ is the level of suboptimality. We shorthand $\iota = \log(SAH^2K^{3/2}(1+\rho))$.

| | Algorithm | Reactangular | Regret | Sample Complexity |
|---|---|---|---|---|
| Wang and Zou [2021] | Value based | $(s,a)$ | NA | Asymptotic |
| Badrinath and Kalathil [2021] | Policy based | $(s,a)$ | NA | Asymptotic |
| **Ours** | Policy based | $(s,a)$ | $O\left(SH^2\sqrt{AK\iota}\right)$ | $O\left(\frac{H^4S^2A\iota}{\epsilon^2}\right)$ |
| | Policy based | s | $O\left(SA^2H^2\sqrt{K\iota}\right)$ | $O\left(\frac{H^4S^2A^4\iota}{\epsilon^2}\right)$ |

setting: the agent sequentially interacts with the environment and tackles the exploration-exploitation challenge as it balances between exploring the state space and exploiting the high-reward actions. A practical motivation for using the robust online MDP formation is as follows: The policy learned through reinforcement learning policy can only be obtained through interacting with a simulator, but ultimately, it is asked to minimize the regret in the real environment. However, the simulator can be inherently inaccurate since it is always just an approximation of the real world. The disparity between the simulation and real environment can be modeled through the online robust MDP setting.

The online setup was well-understood in standard MDP problems Jin et al. [2019], Rosenberg and Mansour [2019], Jin and Luo [2020]. Yet, in the robust MDP setting, previous sample complexity results cannot directly imply a sublinear regret, in general, Dann et al. [2017]. A natural question then arises:

*Can we design a robust RL algorithm that attains sublinear regret under robust MDP with rectangular uncertainty set?*

We propose the first policy optimization algorithm for robust MDP under a rectangular uncertainty set. One of the challenges for deriving a regret guarantee for robust MDP stems from its adversarial nature. As the transition dynamic can be picked adversarially from a predefined set, the optimal policy may be randomized Wiesemann et al. [2013]. This is in contrast with conventional MDPs, where there always exists a deterministic optimal policy, which can be found with value-based methods and a greedy policy (e.g. UCB-VI algorithms). Bearing this observation, we resort to policy optimization (PO)-based methods, which directly optimize a stochastic policy in an incremental way.

With a stochastic policy, our algorithm explores robust MDPs in an optimistic manner. To achieve this robustly, we propose a carefully designed bonus function via the dual conjugate of the robust Bellman equation. This quantifies both the uncertainty stemming from the limited historical data and the uncertainty of the MDP dynamic. In the episodic setting of robust MDPs, we show that our algorithm attains sublinear regret $O(\sqrt{K})$ for both $(s,a)$ and

$s$-rectangular uncertainty set, where $K$ is the number of episodes. In the case where the uncertainty set contains only the nominal transition model, our results recover the previous regret upper bound of non-robust policy optimization Shani et al. [2020]. Our result achieves the first provably efficient regret bound in the online robust MDP problem, as shown in Table 1. We further validated our algorithm with experiments.

## 2 RELATED WORK

**RL with robust MDP** Robust MDPs allow the transition kernel to take values from an uncertainty set. The objective of robust MDPs is to learn an optimal robust policy that maximizes the worst-case value function. When the exact information about MDP is known, this can be solved through dynamic programming methods Iyengar [2005], Nilim and El Ghaoui [2005], Mannor et al. [2012]. If one has access to a generative model, several model-based reinforcement learning methods are proven to be statistically efficient Panaganti and Kalathil [2022], Yang et al. [2021]. Similar results can also be achieved if an offline dataset is present, for which previous works Qi and Liao [2020], Zhou et al. [2021], Kallus et al. [2022], Ma et al. [2022] show the $O(1/\epsilon^2)$ sample complexity for an $\epsilon$-optimal policy. In addition, Liu et al. [2022] proposed distributionally robust policy Q-learning, which solves for the asymptotically optimal Q-function.

In an online setting, the only results available are asymptotic. In the case of discounted MDPs, Wang and Zou [2021], Badrinath and Kalathil [2021] study the policy gradient method and show an $O(\epsilon^{-3})$ convergence rate for an alternative learning objective (a smoothed variant), which could be equivalent to the original policy gradient objective in an asymptotic regime. These results in sample complexity and asymptotic regimes, in general, cannot imply sublinear regret in robust MDPs Dann et al. [2017]. We summarize known results in the online setting in Table 1. We note that value estimation Panaganti and Kalathil [2022], Yang et al. [2021] does not directly lead to an optimal policy but we convert the rates by applying an additional value iteration step.

**RL with adversarial MDP** We differ our problem setup from another framework, often referred to as the adversarial MDP, where the MDP parameters can be adversarially chosen while the agent interacts with the environment. This problem is more challenging than robust MDP because robust MDP assumed that the agent interacts with a fixed environment and is tested on adversarial tasks. In general, adversarial MDP is proved to be NP-hard Even-Dar et al. [2004]. Several works study the variant where the adversarial could only modify the reward function, while the transition dynamics of the MDP remain unchanged. In this case, it is possible to obtain policy-based algorithms that are efficient with a sublinear regret Rosenberg and Mansour [2019], Jin and Luo [2020], Jin et al. [2020], Shani et al. [2020], Cai et al. [2020]. Alternatively, researchers investigate the setting where the transition is only allowed to be adversarially chosen for $C$ out of the $K$ total episodes. A regret of $O(C^2 + \sqrt{K})$ are established thereafter Lykouris et al. [2021], Chen et al. [2021b], Zhang et al. [2022].

**Non-robust policy optimization** Policy optimization has been extensively investigated under non-robust MDPs Neu et al. [2010], Cai et al. [2020], Shani et al. [2020], Wu et al. [2022], Chen et al. [2021a]. The proposed methods are proven to be able to achieve sublinear regret. The methods are also closely related to empirically successful policy optimization algorithms in RL, such as PPO Schulman et al. [2017] and TRPO Schulman et al. [2015].

# 3 ROBUST MDP AND UNCERTAINTY SETS

In this section, we describe the robust MDP and start with defining some notations.

**Robust Markov decision process** We consider an episodic finite horizon tabular robust MDP, which can be denoted by a tuple $\mathcal{M} = \langle \mathcal{S}, \mathcal{A}, H, \{\mathcal{P}_h\}_{h=1}^H, \{r\}_{h=1}^H \rangle$. Here $\mathcal{S}$ is the state space, $\mathcal{A}$ is the action space, $\{r\}_{h=1}^H$ is the time-dependent reward function, and $H$ is the length of each episode. Instead of a fixed uncertainty kernel, the transitions of the robust MDP are governed by kernels from a time-dependent uncertainty set $\{\mathcal{P}_h\}_{h=1}^H$, *i.e.*, time-dependent transition $P_h \in \mathcal{P}_h \subseteq \Delta_{\mathcal{S}}$ at time $h$.

The uncertainty set $\mathcal{P}$ is constructed around a nominal transition kernel $\{P_h^o\}$, and all transition dynamics within the set are close to the nominal kernel with a distance metric of one's choice. Different from an episodic finite-horizon non-robust MDP, the transition kernel $P$ be chosen (even adversarially) from a specified time-dependent uncertainty set $\mathcal{P}$. We consider the case where the rewards are stochastic. This is, on state-action $(s, a)$ at time $h$, the immediate reward is $R_h(s, a) \in [0, 1]$, which is drawn i.i.d from a distribution with expectation $r_h(s, a)$. With the described setup

of robust MDPs, we now define the policy and its associated value.

**Policy and robust value function** A time-dependent policy $\pi$ is defined as $\pi = \{\pi_h\}_{h=1}^H$, where each $\pi_h$ is a function from $\mathcal{S}$ to the probability simplex over actions, $\Delta(\mathcal{A})$. If the transition kernel is fixed to be $P$, the performance of a policy $\pi$ starting from state $s$ at time $h$ can be measured by its value function, which is defined as

$$V_h^{\pi,P}(s) = \mathbb{E}_{\pi,P} \left[ \sum_{h'=h}^H r_{h'}(s_{h'}, a_{h'}) \mid s_h = s \right].$$

In robust MDP, the robust value function instead measures the performance of $\pi$ under the worst possible choice of transition $P$ within the uncertainty set. Specifically, the value and the Q-value function of a policy given the state action pair $(s, a)$ at step $h$ are defined as

$$V_h^\pi(s) = \min_{\{P_h\} \in \{\mathcal{P}_h\}} V_h^{\pi, \{P_h\}}(s),$$

$$Q_h^\pi(s, a) = \min_{\{P_h\} \in \{\mathcal{P}_h\}} \mathbb{E}_{\pi, \{P_h\}} \left[ \sum_{h'=h}^H r_h(s_{h'}, a_{h'}) \mid (s_h, a_h) = (s, a) \right].$$

The optimal value function is defined to be the best possible value attained by a policy $V_h^*(s) = \max_\pi V_h^\pi(s) = \max_\pi \min_{\{P_h\} \in \{\mathcal{P}_h\}} V_h^{\pi, \{P_h\}}(s)$. The optimal policy is then defined to be the policy that attains the optimal value.

**Robust Bellman equation** Similar to non-robust MDP, robust MDP has the following robust bellman equation, which characterizes a relation to the robust value function Wiesemann et al. [2013], Ho et al. [2021], Yang et al. [2021], Behzadian et al. [2021].

$$Q_h^\pi(s, a) = r(s, a) + \sigma_{\mathcal{P}_h}(V_{h+1}^\pi)(s, a),$$
$$V_h^\pi(s) = \langle Q_h^\pi(s, \cdot), \pi_h(\cdot, s) \rangle,$$

where

$$\sigma_{\mathcal{P}_h}(V_{h+1}^\pi)(s, a) = \min_{P_h \in \mathcal{P}_h} P_h(\cdot \mid s, a) V_{h+1}^\pi,$$

$$P_h(\cdot \mid s, a) V = \sum_{s' \in \mathcal{S}} P_h(s' \mid s, a) V(s'). \qquad (1)$$

Without additional assumptions on the uncertainty set, the optimal policy and value of the robust MDP are in general NP-hard to solve Wiesemann et al. [2013]. One of the most common assumptions that make solving optimal value feasible is the rectangular assumption Iyengar [2005], Wiesemann et al. [2013], Badrinath and Kalathil [2021], Yang et al. [2021], Panaganti and Kalathil [2022].

**Rectangular uncertainty sets** To limit the level of perturbations, we assume that the transition kernel is close to the nominal transition measured via $\ell_1$ distance. We consider two cases.

The $(s, a)$-rectangular assumption assumes that the uncertain transition kernel within the set takes value independently for each $(s, a)$. We further use $\ell_1$ distance to characterize the $(s, a)$-rectangular set around a nominal kernel with a specified level of uncertainty.

**Definition 3.1** (($(s, a)$-rectangular uncertainty set Iyengar [2005], Wiesemann et al. [2013]). *For all time step $h$ and with a given state-action pair $(s, a)$, the $(s, a)$-rectangular uncertainty set $\mathcal{P}_h(s, a)$ is defined as*

$$\mathcal{P}_h(s, a) = \{ \|P_h(\cdot \mid s, a) - P_h^o(\cdot \mid s, a)\|_1 \leq \rho, \\ P_h(\cdot \mid s, a) \in \Delta(\mathcal{S})\},$$

*where $P_h^o$ is the nominal transition kernel at $h$, $P_h^o(\cdot \mid s, a) \geq c > 0, \forall (s, a) \in \mathcal{S} \times \mathcal{A}$, $\rho$ is the level of uncertainty and $\Delta(\mathcal{S})$ denotes the probability simplex over the state space $\mathcal{S}$.*

With the $(s, a)$-rectangular set, it is shown that there always exists an optimal policy that is deterministic Wiesemann et al. [2013].

One way to relax the $(s, a)$-rectangular assumption is to instead let the uncertain transition kernels within the set take value independent for each $s$ only. This characterization is then more general and its solution gives a stronger robustness guarantee.

**Definition 3.2** ($s$-rectangular uncertainty set Wiesemann et al. [2013]). *For all time step $h$ and with a given state $s$, the $s$-rectangular uncertainty set $\mathcal{P}_h(s)$ is defined as*

$$\mathcal{P}_h(s) = \left\{ \sum_{a \in \mathcal{A}} \|P_h(\cdot \mid s, a) - P_h^o(\cdot \mid s, a)\|_1 \leq A\rho, \\ P_h(\cdot \mid s, \cdot) \in \Delta(\mathcal{S})^{\mathcal{A}} \right\},$$

*where $P_h^o$ is the nominal transition kernel at $h$, $P_h^o(\cdot \mid s, a) > 0, \forall (s, a) \in \mathcal{S} \times \mathcal{A}$, $\rho$ is the level of uncertainty, and $\Delta(\mathcal{S})$ denotes the probability simplex over the state space $\mathcal{S}$.*

Different from the $(s, a)$-rectangular assumption, which guarantees the existence of a deterministic optimal policy, the optimal policy under $s$-rectangular set may need to be randomized Wiesemann et al. [2013]. We also remark that the requirement of $P_h^o(\cdot \mid s, a) > 0$ is mostly for technical convenience.

Equipped with the characterization of the uncertainty set, we now describe the learning protocols and the definition of regret under the robust MDP.

**Learning protocols and regret** We consider a learning agent repeatedly interacts with the environment in an episodic manner, over $K$ episodes. At the start of each episode, the learning agent picks a policy $\pi_k$ and interacts with the environment while executing $\pi_k$. Without loss of generality, we assume the agents always start from a fixed initial state $s$. The performance of the learning agent is measured by the cumulative regret incurred over the $K$ episodes. Under the robust MDP, the cumulative regret is defined to be the cumulative difference between the robust value of $\pi_k$ and the robust value of the optimal policy, $\text{Regret}(K) = \sum_{k=1}^{K} V_1^*(s_1^k) - V_1^{\pi_k}(s_1^k)$, where $s_1^k$ is the initial state in episode $k$.

We highlight that the transition of the states in the learning process is specified by the nominal transition kernel $\{P_h^o\}_{h=1}^{H}$, though the agent only has access to the nominal kernel in an online manner. We remark that if the agent is asked to interact with a potentially adversarially chosen transition from an arbitrary uncertainty set, the learning problem is NP-hard Even-Dar et al. [2004].

One practical motivation for this formulation could be as follows. The policy provider only sees feedback from the nominal system, yet it aims to minimize the regret for clients who refuse to share additional deployment details for purposes such as privacy concerns. Thus the observed feedback describes the "nominal transition" while the unseen clients are represented by the "uncertainty set".

# 4 ALGORITHM

Before we introduce our algorithm, we first illustrate the importance of taking uncertainty into consideration. With the robust MDP, one of the most naive methods is to train a policy directly with the nominal transition model. However, the following proposition shows an optimal policy under the nominal policy can be arbitrarily bad in the worst-case transition (even worse than a random policy).

**Claim 4.1** (Suboptimality of non-robust optimal policy). *There exists a robust MDP $\mathcal{M} = \langle \mathcal{S}, \mathcal{A}, \mathcal{P}, r, H \rangle$ with uncertainty set $\mathcal{P}$ of uncertainty radius $\rho$, such that the non-robust optimal policy is $\Omega(1)$-suboptimal to the uniformly random policy.*

The proof of Proposition 4.1 is deferred to Appendix D. This result is obviously not ideal, and it motivates us to to propose an algorithm that works well even when the models mismatch. Indeed, we present below the robust optimistic policy optimization (Algorithm 1), which enjoys a sublinear regret and desired practical performance.

## 4.1 ROBUST OPTIMISTIC POLICY OPTIMIZATION

With the presence of the uncertainty set, the optimal policies may be all randomized [Wiesemann et al., 2013]. In such cases, value-based methods may be insufficient as they

**Algorithm 1** Robust Optimistic Policy Optimization (ROPO)

Input: learning rate $\beta$, bonus function $b_h^k$.
**for** $k = 1, \ldots, K$ **do**
  Collect a trajectory of samples by executing $\pi_k$.
  **for** $h = H, \ldots, 1$ **do**
    **for** $\forall (s, a) \in \mathcal{S} \times \mathcal{A}$ **do**
      Solve $\sigma_{\hat{\mathcal{P}}_h}(\hat{V}_{h+1}^k)(s, a)$, according to Equation (3) for $(s, a)$-rectangular set or Equation (4) for $s$-rectangular set.
      $$\hat{Q}_h^k(s, a) = \min\left\{\hat{r}(s, a) + \sigma_{\hat{\mathcal{P}}_h}(\hat{V}_{h+1}^k)(s, a) + b_h^k(s, a), H\right\}.$$
    **end for**
    **for** $\forall s \in \mathcal{S}$ **do**
      $\hat{V}_h^k(s) = \left\langle \hat{Q}_h^k(s, \cdot), \pi_h^k(\cdot \mid s) \right\rangle$.
    **end for**
  **end for**
  $\pi_h^{k+1}(a \mid s) = \frac{\pi_h^k(a|s) \exp(\beta \hat{Q}_h^\pi(s,a))}{\sum_{a'} \pi_h^k(a'|s) \exp(\beta \hat{Q}_h^\pi(s,a'))}, \forall h, s, a \in [H] \times \mathcal{S} \times \mathcal{A}$
  Update empirical estimate $\hat{r}$, $\hat{P}$ with Equation (2).
**end for**

---

usually rely on a deterministic policy. We thus resort to optimistic policy optimization methods Shani et al. [2020], which directly learn a stochastic policy.

Our algorithm performs policy optimization with empirical estimates and encourages exploration by adding a bonus to less explored states. However, we need to propose a new efficiently computable bonus that is robust to adversarial transitions. We achieve this by solving a sub-optimization problem derived from Fenchel conjugate. We present Robust Optimistic Policy Optimization (ROPO) in Algorithm 1 and elaborate on its design components.

**The empirical model** To start, as our algorithm has no access to the actual reward and transition function, we use the following empirical estimator of the transition and reward:

$$\hat{r}_h^k(s, a) = \frac{\sum_{k'=1}^{k-1} R_h^{k'}(s, a) \mathbb{I}_{s_h^{k'}, a_h^{k'}}^{s,a}}{N_h^k(s, a)},$$

$$\hat{P}_h^{o,k}(s, a, s') = \frac{\sum_{k'=1}^{k-1} \mathbb{I}_{s_h^{k'}, a_h^{k'}, s_{h+1}^{k'}}^{s,a,s'}}{N_h^k(s, a)}, \quad (2)$$

where

$$\mathbb{I}_{s_h^{k'}, a_h^{k'}}^{s,a} = \mathbb{I}\left\{\left(s_h^{k'}, a_h^{k'}\right) = (s, a)\right\}$$

$$N_h^k(s, a) = \max\left\{\sum_{k'=1}^{k-1} \mathbb{I}_{s_h^{k'}, a_h^{k'}}^{s,a}, 1\right\}$$

counts the number of visits to $(s, a)$.

**Challenge: Optimistic robust policy evaluation** As in standard optimistic algorithms, Algorithm 1 estimates $Q$-values with an optimistic variant of the Bellman equation to encourage exploration in the robust MDP. The bonus term $b_h^k(s, a)$ compensates for the lack of knowledge of the actual reward and transition model as well as the uncertainly set, with order $b_h^k(s, a) = O(N_h^k(s, a)^{-1/2})$.

However, in the robust MDP setting, analyzing the bonus term can be tricky. Intuitively, the bonus term $b_h^k$ desires to characterize the optimism required for efficient exploration for both the estimation errors of $P$ and the robustness of $P$. It is hard to control the two quantities in their primal (original) form because it is unclear how the error in estimating $P$ would impact the choice of an estimated robust action $\sigma_{\hat{\mathcal{P}}_h}$.

We propose the following procedure to address the problem. Note that the key difference between our algorithm and standard policy optimization is that $\sigma_{\hat{\mathcal{P}}_h}(\hat{V}_{h+1}^\pi)(s)$ requires solving an inner minimization (1). Through relaxing the constraints with Lagrangian multiplier and Fenchel conjugates, under $(s, a)$-rectangular set, the inner minimization problem can be reduced to a one-dimensional unconstrained convex optimization problem on $\mathbb{R}$ (Lemma A.4).

$$\sup_\eta \eta - \frac{(\eta - \min_s \hat{V}_{h+1}^{\pi_k}(s))_+}{2}\rho$$
$$- \sum_{s'} \hat{P}_h^o(s' \mid s, a)\left(\eta - \hat{V}_{h+1}^{\pi_k}(s')\right)_+. \quad (3)$$

The optimum of Equation (3) can be computed efficiently with bisection or sub-gradient methods. More importantly, this form allows us to estimate how the error of estimating the transition kernel impact the estimated value function while bypassing $\sigma_{\hat{\mathcal{P}}_h}$.

Similarly, in the case of $s$-rectangular set, the inner minimization problem is equivalent to a $A$-dimensional convex optimization problem.

$$\sup_\eta \sum_{a'} \eta_{a'} - \sum_{s',a'} \hat{P}_h^o(s' \mid s, a')\left(\eta_{a'} - \mathbb{I}\{a' = a\}\hat{V}_{h+1}^{\pi_k}(s')\right)_+$$
$$- \min_{s',a'} \frac{A\rho(\eta_{a'} - \mathbb{I}\{a' = a\}\hat{V}_{h+1}^{\pi_k}(s'))_+}{2}, \quad (4)$$

where $a \sim \pi_k(s)$.

In addition to reducing computational complexity, the dual form (Equation (3) and Equation (4)) decouples the uncertainty in estimation error and in robustness, as $\rho$ and $\hat{P}_h^o$ are in different terms. The exact form of $b_h^k$ is presented in the Equation (5) and (6).

**Policy improvement step** Using the optimistic $Q$-value obtained from policy evaluation, the algorithm improves the policy with a KL regularized online mirror descent step, $\pi_h^{k+1} \in \arg\max_\pi \beta\langle \nabla \hat{V}_h^{\pi_k}, \pi \rangle - \pi_h^k + D_{KL}(\pi \| \pi_h^k)$,

where $\beta$ is the learning rate. Equivalently, the updated policy is given by the closed-form solution $\pi_h^{k+1}(a \mid s) = \frac{\pi_h^k \exp(\beta \hat{Q}_h^\pi(s,a))}{\sum_{a'} \pi_h^k(a'|s) \exp(\beta \hat{Q}_h^\pi(s,a'))}$. An important property of policy improvement is to use a fundamental inequality (7) of online mirror descent presented in [Shani et al., 2020]. We suspect that other online algorithms with sublinear regret could also be used in policy improvement.

In the non-robust case, this improvement step is also shown to be theoretically efficient [Shani et al., 2020, Wu et al., 2022]. Many empirically successful policy optimization algorithms, such as PPO [Schulman et al., 2017] and TRPO Schulman et al. [2015], also take a similar approach to KL regularization for non-robust policy improvement. Putting everything together, the proposed algorithm is summarized in Algorithm 1.

# 5 THEORETICAL RESULTS

We are now ready to analyze the theoretical results of our algorithm under the uncertainly set.

## 5.1 RESULTS UNDER $(s, a)$-RECTANGULAR UNCERTAINTY SET

Equipped with Algorithm 1 and the bonus function described in Equation (5). We obtain the regret upper bound under $(s, a)$-rectangular uncertainty set described in the following theorem.

**Theorem 5.1.** *With learning rate $\beta = \sqrt{\frac{2 \log A}{H^2 K}}$ and bonus term $b_h^k$ as (5), with probability at least $1 - \delta$, the regret incurred by Algorithm 1 over $K$ episodes is bounded by*

$$Regret(K) = O\left(\frac{H^2 S}{c} \sqrt{AK \log\left(SAH^2 K^{3/2}(1 + \rho)/\delta\right)}\right).$$

By the definition of $\rho$ as the $\ell_1$ distance, it is at most 1. This indicates that the regret scales logarithmically with $\rho$ for small $\rho$, and is capped at some constant in the large-$\rho$ regime. It concludes that our algorithm derives a robust enough policy in a way that if there is a policy that achieves a high return then this policy achieves a high return.

**Remark 5.1.** *When $\rho = 0$, the problem reduces to non-robust reinforcement learning. In such case, our regret upper bound is $\tilde{O}\left(H^2 S \sqrt{AK}/c\right)$, which is $1/c$ order away from the regret bound for policy optimization algorithms for the non-robust case Shani et al. [2020]. This is due to the over-cautiousness of our algorithm for robust performance. Our algorithm subtly characterizes the uncertainty from all sources, it derives a robust enough policy in a way that if there is a policy that achieves a high return then this policy achieves a high return. Thus when $\rho > 0$, our result is*

*only logarithmically dependent on $\rho$. We further note that this also matches the dependency in the sample complexity results Yang et al. [2021], Panaganti and Kalathil [2022].*

While we defer the detailed proof to Appendix A, we highlight the challenges in the proof below.

We start with decomposing the regret as

$$\sum_{k=1}^K (V_1^*(s) - \hat{V}_1^{\pi_k}(s)) + \sum_{k=1}^K (\hat{V}_1^{\pi_k}(s) - V_1^{\pi_k}(s)).$$

In the case of policy optimization for non-robust MDP, the first term is upper bounded through the value difference lemma [Shani et al., 2020]. Yet this can be no longer applied to the robust MDP case, due to the presence of policy-dependent adversarial transition kernel. Moreover, naively employing a recursive relation with respect to a fixed transition kernel in a similar way to the value difference lemma may lead to linear regret.

To address the issue of varying transition kernel, we decompose the first term as,

$$V_h^*(s) - \hat{V}_h^{\pi_k}(s)$$
$$\leq \langle \hat{Q}_h^{\pi_k}(s, \cdot), \pi_*(\cdot \mid s) - \pi_k(\cdot \mid s) \rangle + \mathbb{E}_{\pi_*} \Big[ (r_h(s, a) - \hat{r}_h^k(s, a))$$
$$\quad + (\sigma_{\mathcal{P}_h(s,a)}(\hat{V}_{h+1}^{\pi_k})(s, a) - \sigma_{\hat{\mathcal{P}}_h(s,a)}(\hat{V}_{h+1}^{\pi_k})(s, a) - b_h^k(s, a) \Big]$$
$$\quad + \mathbb{E}_{\pi_*} \Big[ \sigma_{\mathcal{P}_h(s,a)}(V_{h+1}^*)(s, a) - \sigma_{\mathcal{P}_h(s,a)}(\hat{V}_{h+1}^{\pi_k})(s, a) \Big].$$

We then apply this decomposition repeatedly by conditioning on varying transition kernel $q_h(\cdot \mid s, a) = \arg\max_{P_h \in \mathcal{P}_h} P_h(\cdot \mid s, a)(\hat{V}_{h+1}^{\pi_k} - V_{h+1}^{\pi_k})$. By setting the optimism bonus $b_h^k(s, a)$ carefully, we obtain

$$\sum_{k=1}^K V_1^*(s) - \hat{V}_1^{\pi_k}(s)$$
$$\leq \sum_{k=1}^K \sum_{h=1}^H \mathbb{E}_{\pi_*, \{q_t\}_{t=1}^{h-1}} \Big[ \langle \hat{Q}_h^{\pi_k}(s, \cdot), \pi_*(\cdot \mid s) - \pi_k(\cdot \mid s) \rangle \Big],$$

This can be upper bounded by standard results of online mirror descent.

However, we remark that designing such a bonus function is non-trivial as the expectation of each time steps $h$ is taken with respect to a different transition kernel. To establish such optimism bonus, we first derive the dual formulation of inner optimization problem $\sigma_{\hat{\mathcal{P}}_{(s,a)}}(V)$ (Equation (3)). This allows us to decouple the uncertainty and bound each source of uncertainty separately. With a change of variable $\tilde{P}_h(s' \mid s, a) = \frac{P_h(s'|s,a)}{P_h^o(s'|s,a)}$, we can write the Lagrangian

form of $\sigma_{\mathcal{P}_h(s,a)}(\hat{V}_{h+1}^{\pi_k})(s,a)$ as

$$\sum_{s'} \tilde{P}_h(s' \mid s,a) P_h^o(s' \mid s,a) \hat{V}_{h+1}^{\pi_k}(s')$$

$$+ \lambda \left( \sum_{s'} |\tilde{P}_h(s' \mid s,a) - 1| P_h^o(s' \mid s,a) - \rho \right)$$

$$- \eta \left( \sum_{s'} \tilde{P}_h(s' \mid s,a) P_h^o(s' \mid s,a) - 1 \right),$$

where $\eta, \lambda$ are both Lagrangian multipliers.

Under the characterization of $\ell_1$ distance, we can use the convex conjugate of $f(x) = |x - 1|$ to optimize out $\tilde{P}$, resulting with Equation (3). Notice that now the difference of $\sigma_{\hat{\mathcal{P}}_{(s,a)}}(V) - \sigma_{\mathcal{P}_{(s,a)}}(V)$ is only incurred by the difference in $\sum_{s'} P_h^o(s' \mid s,a) \left( \eta - \hat{V}_{h+1}^{\pi_k}(s') \right)_+$. We then show that $\eta$ must be bounded at its optimum by inspecting certain pivot points and by the convexity of the dual. When we have the desired bounds of $\eta$, applying Hoeffding's inequality with an $\epsilon$-net argument will yield the desired bonus function.

Our algorithm and analysis techniques can also extend to other probability distances, such as KL divergence constrained uncertainly set. We include the result for KL divergence in Appendix C.

## 5.2 RESULTS UNDER $s$-RECTANGULAR UNCERTAINTY SET

Beyond the $(s,a)$-rectangular uncertainty set, we also extend to $s$-rectangular uncertainty set (Definition 3.2). Recall that value-based methods do not extend to $s$-rectangular uncertainty set as there might not exist a deterministic optimal policy.

**Theorem 5.2** (Regret under $s$-rectangular uncertainty set). *With learning rate $\beta = \sqrt{\frac{2 \log A}{H^2 K}}$ and bonus term $b_h^k$ as (6), with probability at least $1 - \delta$, the regret of Algorithm 1 is bounded by*

$$Regret(K) = O\left( \frac{SA^2 H^2}{c} \sqrt{K \log(SA^2 H^2 K^{3/2}(1+\rho)/\delta)} \right)$$

**Remark 5.2.** *When $\rho = 0$, the problem reduces to non-robust reinforcement learning. In such case, our regret upper bound is $\tilde{O}\left( SA^2 H^2 \sqrt{K}/c \right)$. Our result is the first theoretical result for learning a robust policy under $s$-rectangular uncertainty set, as previous results only learn the robust value function [Yang et al., 2021]. When $\rho > 0$, our result is only logarithmically dependent on $\rho$, which matches the dependency in the sample complexity results Yang et al. [2021], Panaganti and Kalathil [2022].*

The analysis and techniques used for Theorem 5.2 hold great similarity to those ones used for Theorem 5.1. The main difference is on bounding $\sigma_{\hat{\mathcal{P}}_h(s)}(\hat{V}_{h+1}^{\pi_k})(s,a) - \sigma_{\mathcal{P}_h(s)}(\hat{V}_{h+1}^{\pi_k})(s,a)$. We defer the detailed proof to the appendix B.

## 6 EMPIRICAL RESULTS

To validate our theoretical findings, we conduct a preliminary empirical analysis of our purposed robust policy optimization algorithm. We are committed to making our implementation public.

| Start | o | o | o | o |
|---|---|---|---|---|
| o | x | o | o | o |
| o | o | x | o | o |
| o | o | o | x | o |
| o | o | o | o | + |

Figure 1: Example of the Gridworld environment.

**Environment** We conduct the experiments with the Gridworld environment, which is an early example of reinforcement learning Sutton and Barto [2018]. The environment is two-dimensional and is in a cell-like environment. Specifically, the environment is a $5 \times 5$ grid, where the agent starts from the upper left cell. The cells consist of three types, road (labeled with $\circ$), wall (labeled with $\times$), and the reward state (labeled with $+$).

The agent can walk through the road cell but not the wall cell. If it attempts to move to a wall cell, it will not move. Once the agent steps on the reward cell, it will receive a reward of 1, and it will receive no rewards otherwise. The goal of the agents is to collect as many rewards as possible within the allowed time. The agent has four types of actions at each step, up, down, left, and right. After taking the action, the agent has a success probability of $p$ to move according to the desired direction, and with the remaining probability of moving to other directions uniformly randomly. Figure 1 shows an example of our environment.

**Experiment configurations** To simulate the robust MDP, we create a nominal transition dynamic with move success probability $p = 0.9$. The learning agent will interact with this nominal transition during training time and interact with a perturbed transition dynamic during evaluation. Under $(s,a)$-rectangular set, the transitions are perturbed against the direction the agent is directing with a constraint of $\rho$. Under $s$-rectangular set, the transitions are perturbed against the direction of the goal state. For example, if the agent chooses to go down to reach the goal state, the perturbation will be against the agent's direction (upward) by $\rho$. This adversarial change of transition is an implementation of the adversarial behavior described by robust MDP $\min_{\{P_h\} \in \{\mathcal{P}_h\}} V_h^{\pi, \{P_h\}}(s)$. It is obvious that the perturbation caused some of the optimal policies under nominal transition to be sub-optimal under robust transitions.

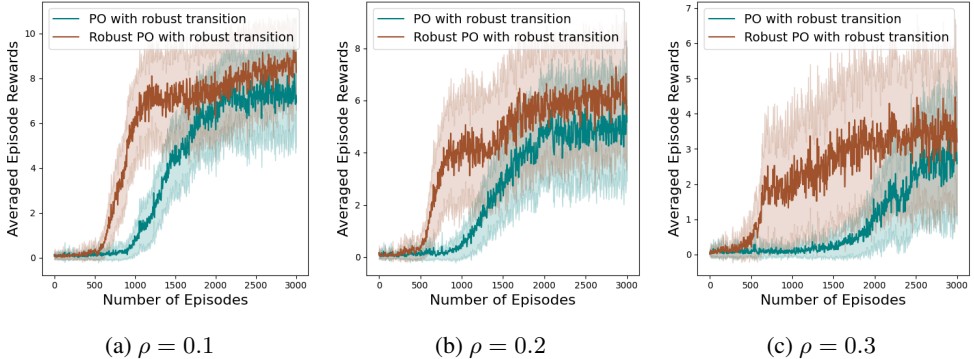

Figure 2: Cumulative rewards obtained by robust and non-robust policy optimization on robust transition with different levels of uncertainty $\rho = 0.1, 0.2, 0.3$ under $\ell_1$ distance, $(s, a)$-rectangular set.

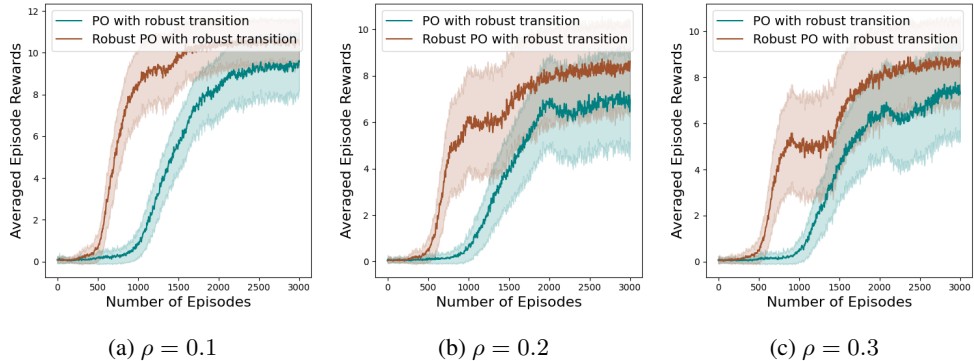

Figure 3: Cumulative rewards obtained by robust and non-robust policy optimization on robust transition with different levels of uncertainty $\rho = 0.1, 0.2, 0.3$ under $\ell_1$ distance, $s$-rectangular set.

**Results** We denote the perturbed transition as robust transitions in our results. We implement our proposed robust policy optimization algorithm along with the non-robust variant of it Shani et al. [2020]. The inner minimization of our Algorithm 1 is computed through its dual formulation for efficiency. Our algorithm is implemented with the RLberry framework [Domingues et al., 2021].

We present results with $\rho = 0.1, 0.2, 0.3$ under $(s, a)$-rectangular set here in Figure 2, and under $s$-rectangular set here in Figure 3. We present the averaged cumulative rewards during evaluation. Regardless of the level of uncertainty and choice of uncertainty set, we observe that the robust variant of the policy optimization algorithm is more robust to dynamic changes as it is able to obtain a higher level of rewards than its non-robust variant.

## 7 CONCLUSION AND FUTURE DIRECTIONS

In this paper, we studied the problem of regret minimization in robust MDP with a rectangular uncertainty set. We proposed a robust variant of optimistic policy optimization, which achieves sublinear regret in all uncertainty sets considered. Our algorithm delicately balances the exploration-exploitation trade-off through a carefully designed bonus term, which quantifies not only the uncertainty due to the limited observations but also the uncertainty of robust MDPs. Our results are the first regret upper bounds the first non-asymptotic results in robust MDPs, without access to a generative model.

For future works, while our analysis achieves the same bound as the policy optimization algorithm in Shani et al. [2020] when the robustness level $\rho = 0$, we suspect some technical details could be improved. For instance, we needed $P_h^o$ to be positive for any $s, a$ to form a solvable Fenchel dual. However, this positive value is canceled later and does not appear in the bound. This suggests that the strictly positive assumption may be an analysis artifact. Additionally, we can explore other uncertainty set characterizations, such as the Wasserstein distance metric. We can also extend robust MDPs to a broader class of MDPs, such as those with infinitely many states and function approximation.

## ACKNOWLEDGEMENT

Jing Dong and Baoxiang Wang are partially supported by the National Natural Science Foundation of China (62106213, 72394361).

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

# Online Policy Optimization for Robust Markov Decision Process
## (Supplementary Material)

**Jing Dong** [*1]  **Jingwei Li** [*2]  **Baoxiang Wang** [*1,3]  **Jingzhao Zhang** [*2,4]

[1]The Chinese University of Hong Kong, Shenzhen
[2]Institute for Interdisciplinary Information Sciences, Tsinghua University
[3]Vector Institute
[4]Shanghai Qizhi Institute

## A  PROOFS OF THEOREM 5.1

### A.1  GOOD EVENTS

We first define the following good events, in which case we estimate the reward function and the nominal transition functions fairly accurately.

$$\mathcal{G}_k^r = \left\{ \forall s, a, h : \left| r_h(s,a) - \hat{r}_h^k(s,a) \right| \leq \sqrt{\frac{2 \ln(2SAH^2K/\delta')}{N_h^k(s,a)}} \right\},$$

$$\mathcal{G}_k^p = \left\{ \forall s, a, h : \sigma_{\mathcal{P}_h(s,a)}(\hat{V}_{h+1}^{\pi_k})(s,a) - \sigma_{\hat{\mathcal{P}}_h(s,a)}(\hat{V}_{h+1}^{\pi_k})(s,a) \leq C_h^k(s,a) \right\},$$

where $C_h^k(s,a) = H\sqrt{\frac{4S \log(3SAH^2K^{3/2}(4+\rho)/\delta')}{N_h^k(s,a)}} + \frac{1}{\sqrt{K}}$.

When the two good events happens at the same time, we say the algorithm in inside the good event $\mathcal{G} = \left( \bigcap_{k=1}^K \mathcal{G}_k^r \right) \bigcap \left( \bigcap_{k=1}^K \mathcal{G}_k^p \right)$. The following lemma shows that $\mathcal{G}$ happens with high probability by setting $\delta'$ properly.

**Lemma A.1** (Good event). *Let $\delta = 2\delta'$, then the good event happens with high probability, i.e. $\mathbb{P}\left[\mathcal{G}\right] \geq 1 - \delta$.*

*Proof.* By Hoeffding's inequality and an union bound on all $s, a$, all possible values of $N_k(s,a)$ and $k$, we have $\mathbb{P}\left[\bigcap_{k=1}^K \mathcal{G}_k^r\right] \geq 1 - \delta'$. By Lemma A.4, we have $\mathbb{P}\left[\bigcap_{k=1}^K \mathcal{G}_k^p\right] \geq 1 - \delta'$ Then set $\delta = 2\delta'$ and we have the desired result. □

### A.2  DESIGN OF THE BONUS FUNCTION

In the case of $(s,a)$-rectangular uncertainty set, we use the following bonus function $b_h^k(s,a)$ to encourage exploration.

$$b_h^k(s,a) = \sqrt{\frac{2 \log(3SAH^2K/\delta)}{N_h^k(s,a)}} + H\sqrt{\frac{4S \log(3SAH^2K^{3/2}(4+\rho)/\delta)}{N_h^k(s,a)}} + \frac{1}{\sqrt{K}}. \tag{5}$$

---

[*]Authors are ordered alphabetically. Corresponding to: Jing Dong, Jingwei Li, Baoxiang Wang, Jingzhao Zhang.
[*]Authors are ordered alphabetically. Corresponding to: Jing Dong, Jingwei Li, Baoxiang Wang, Jingzhao Zhang.

*Accepted for the 40[th] Conference on Uncertainty in Artificial Intelligence* (UAI 2024).

## A.3 REGRET ANALYSIS

Armed with the defined good event, we are now ready to present the analysis of Theorem 5.1, which establishes the regret of the Algorithm under $(s, a)$-uncertainty set.

*Proof.* We start with decomposing the regret as follows,

$$\text{Regret}(K) = \sum_{k=1}^{K} V_1^*(s) - V_1^{\pi_k}(s)$$

$$= \sum_{k=1}^{K} \left( V_1^*(s) - \hat{V}_1^{\pi_k}(s) \right) + \left( \hat{V}_1^{\pi_k}(s) - V_1^{\pi_k}(s) \right) .$$

By Lemma A.2 and Lemma A.3, with probability at least $1 - \delta$, we have

$$\text{Regret}(K) = O\left( \frac{H^2\sqrt{K \log A}}{c} \right) + O\left( \frac{H^2 S}{c} \sqrt{AK \log\left( SAH^2 K^{3/2}(1 + \rho)/\delta \right)} \right)$$

$$= O\left( \frac{H^2 S}{c} \sqrt{AK \log\left( SAH^2 K^{3/2}(1 + \rho)/\delta \right)} \right) .$$

$\square$

**Lemma A.2.** *With probability at least $1 - \delta$, we have*

$$\sum_{k=1}^{K} V_1^*(s) - \hat{V}_1^{\pi_k}(s) = O\left( \frac{H^2\sqrt{K \log A}}{c} \right) .$$

*Proof.* For any $h \in [1, H]$, we have

$$V_h^*(s) - \hat{V}_h^{\pi_k}(s)$$

$$= \langle Q_h^*(s, \cdot), \pi_*(\cdot \mid s) \rangle - \langle \hat{Q}_h^{\pi_k}(s, \cdot), \pi_k(\cdot \mid s) \rangle$$

$$= \langle Q_h^*(s, \cdot) - \hat{Q}_h^{\pi_k}(s, \cdot), \pi_*(\cdot \mid s) \rangle + \langle \hat{Q}_h^{\pi_k}(s, \cdot), \pi_*(\cdot \mid s) - \pi_k(\cdot \mid s) \rangle$$

$$= \mathbb{E}_{\pi_*} \left[ (r_h(s, a) - \hat{r}_h^k(s, a)) + (\sigma_{\mathcal{P}_h(s,a)}(V_{h+1}^*)(s, a) - \sigma_{\hat{\mathcal{P}}_h(s,a)}(\hat{V}_{h+1}^{\pi_k})(s, a)) - b_h^k(s, a) \right]$$

$$\quad + \langle \hat{Q}_h^{\pi_k}(s, \cdot), \pi_*(\cdot \mid s) - \pi_k(\cdot \mid s) \rangle$$

$$= \mathbb{E}_{\pi_*} \left[ (r_h(s, a) - \hat{r}_h^k(s, a)) + (\sigma_{\mathcal{P}_h(s,a)}(\hat{V}_{h+1}^{\pi_k})(s, a) - \sigma_{\hat{\mathcal{P}}_h(s,a)}(\hat{V}_{h+1}^{\pi_k})(s, a)) - b_h^k(s, a) \right]$$

$$\quad + \mathbb{E}_{\pi_*} \left[ \sigma_{\mathcal{P}_h(s,a)}(V_{h+1}^*)(s, a) - \sigma_{\mathcal{P}_h(s,a)}(\hat{V}_{h+1}^{\pi_k})(s, a) \right] + \langle \hat{Q}_h^{\pi_k}(s, \cdot), \pi_*(\cdot \mid s) - \pi_k(\cdot \mid s) \rangle ,$$

where the third equality is by the update rule of our algorithm and the robust bellman equation.

By the design of our bonus function, conditioned on the good event, we have

$$(r_h(s, a) - \hat{r}_h^k(s, a)) + (\sigma_{\mathcal{P}_h(s,a)}(V_{h+1}^*)(s, a) - \sigma_{\hat{\mathcal{P}}_h(s,a)}(\hat{V}_{h+1}^{\pi_k})(s, a)) - b_h^k(s, a) \leq 0 .$$

Let $q_h(\cdot \mid s, a) = \underset{P_h \in \mathcal{P}_h}{\arg\min} \, P_h(\cdot \mid s, a)\hat{V}_{h+1}^{\pi_k}$, then we have

$$\sigma_{\mathcal{P}_h(s,a)}(V_{h+1}^*)(s, a) - \sigma_{\mathcal{P}_h(s,a)}(\hat{V}_{h+1}^{\pi_k})(s, a)$$

$$= \min_{P_h \in \mathcal{P}_h} P_h(\cdot \mid s, a)V_{h+1}^* - \min_{P_h \in \mathcal{P}_h} P_h(\cdot \mid s, a)\hat{V}_{h+1}^{\pi_k}$$

$$= \min_{P_h \in \mathcal{P}_h} P_h(\cdot \mid s, a)V_{h+1}^* - q_h(\cdot \mid s, a)\hat{V}_{h+1}^{\pi_k}$$

$$\leq q_h(\cdot \mid s, a)(V_{h+1}^* - \hat{V}_{h+1}^{\pi_k})$$

$$\leq \max_{P_h \in \mathcal{P}_h} P_h(\cdot \mid s, a)(V_{h+1}^* - \hat{V}_{h+1}^{\pi_k}) .$$

Let $q_h(\cdot \mid s, a) = \underset{P_h \in \mathcal{P}_h}{\arg\max}\, P_h(\cdot \mid s, a)\left((V_{h+1}^*)(s, a) - \hat{V}_{h+1}^{\pi_k}\right)$, Then we have the following relation hold conditioned on the good event:

$$
\begin{aligned}
&V_h^*(s) - \hat{V}_h^{\pi_k}(s) \\
&\le \mathbb{E}_{\pi_*}\left[\sup_{P_h \in \mathcal{P}_h} P_h(\cdot \mid s, a)(V_{h+1}^* - \hat{V}_{h+1}^{\pi_k})\right] + \langle \hat{Q}_h^{\pi_k}(s, \cdot), \pi_*(\cdot \mid s) - \pi_k(\cdot \mid s)\rangle \\
&= \mathbb{E}_{\pi_*, q_h}\left[V_{h+1}^*(s) - \hat{V}_{h+1}^{\pi_k}(s)\right] + \langle \hat{Q}_h^{\pi_k}(s, \cdot), \pi_*(\cdot \mid s) - \pi_k(\cdot \mid s)\rangle.
\end{aligned}
$$

Then, by applying the above relation recursively and with the fact that for any policy $\pi$ and state $s$, $V_{H+1}^*(s) = \hat{V}_{H+1}^{\pi_k}(s) = 0$, we have

$$
V_1^*(s) - \hat{V}_1^{\pi_k}(s) \le \sum_{h=1}^{H} \mathbb{E}_{\pi_*, \{q_t\}_{t=1}^{h-1}}\left[\langle \hat{Q}_h^{\pi_k}(s, \cdot), \pi_*(\cdot \mid s) - \pi_k(\cdot \mid s)\rangle\right].
$$

Let $\omega_h = \{p_t\}_{t=1}^h / P_h^o$, and assume $P_h^o(\cdot \mid s, a) \ge c, \forall (s, a)$. Summing over $k$, we get

$$
\begin{aligned}
\sum_{k=1}^{K} V_1^*(s) - \hat{V}_1^{\pi_k}(s) &\le \sum_{k=1}^{K}\sum_{h=1}^{H} \mathbb{E}_{\pi_*, \{q_t\}_{t=1}^{h-1}}\left[\langle \hat{Q}_h^{\pi_k}(s, \cdot), \pi_*(\cdot \mid s) - \pi_k(\cdot \mid s)\rangle\right] \\
&\le \sum_{k=1}^{K}\sum_{h=1}^{H} \mathbb{E}_{\pi_*, P_h^o}\left[\omega_h \langle \hat{Q}_h^{\pi_k}(s, \cdot), \pi_*(\cdot \mid s) - \pi_k(\cdot \mid s)\rangle\right] \\
&\le \frac{1}{c}\sum_{h=1}^{H} \mathbb{E}_{\pi_*, P_h^o}\left[\sum_{k=1}^{K}\langle \hat{Q}_h^{\pi_k}(s, \cdot), \pi_*(\cdot \mid s) - \pi_k(\cdot \mid s)\rangle\right].
\end{aligned}
$$

By standard results for online mirror descent (Lemma E.3), we have

$$
\sum_{k=1}^{K}\langle \hat{Q}_h^{\pi_k}(s, \cdot), \pi_*(\cdot \mid s) - \pi_k(\cdot \mid s)\rangle \le \frac{\log(A)}{\beta} + \frac{\beta}{2}\sum_{k=1}^{K}\sum_{a \in \mathcal{A}} \pi_h^*(a \mid s)(\hat{Q}_h^{\pi_k}(s, a))^2.
$$

By the update rule of Algorithm 1, we have $0 \le \hat{Q}_h^{\pi_k}(s, a) \le H$, for all $h, k$. Then take $\beta = \sqrt{\frac{2\log A}{H^2 K}}$,

$$
\sum_{k=1}^{K}\langle \hat{Q}_h^{\pi_k}(s, \cdot), \pi_*(\cdot \mid s) - \pi_k(\cdot \mid s)\rangle \le \sqrt{2H^2 K \log A}.
$$

Finally, we have

$$
\sum_{k=1}^{K} V_1^*(s) - \hat{V}_1^{\pi_k}(s) \le \frac{H}{c}\sqrt{2H^2 K \log A} = O\left(\frac{H^2 \sqrt{K \log A}}{c}\right).
$$

$\square$

**Lemma A.3.** *With probability at least $1 - \delta$, we have*

$$
\sum_{k=1}^{K}(\hat{V}_1^{\pi_k} - V_1^{\pi_k})(s) = O\left(\frac{H^2 S}{c}\sqrt{AK \log\left(SAH^2 K^{3/2}(1 + \rho)/\delta\right)}\right).
$$

*Proof.* By the algorithm's update rule and the robust bellman equation, we have

$$(\hat{V}_h^{\pi_k} - V_h^{\pi_k})(s) = \langle \hat{Q}_h^{\pi_k}(s, \cdot) - Q_h^{\pi_k}(s, \cdot), \pi_k(\cdot \mid s)\rangle$$

$$= \left\langle \hat{r}_h^k(s, \cdot) - r_h^k(s, \cdot) + (\sigma_{\hat{\mathcal{P}}_{(s,\cdot)}}(\hat{V}_{h+1}^{\pi_k})(s, \cdot) - \sigma_{\mathcal{P}_{(s,\cdot)}}(V_{h+1}^{\pi_k})(s, \cdot)) + b_h^k(s, \cdot), \pi_k(\cdot \mid s) \right\rangle$$

$$= \mathbb{E}_{\pi_k}\left[\hat{r}_h^k(s, a) - r_h^k(s, a) + (\sigma_{\hat{\mathcal{P}}_h(s,a)}(\hat{V}_{h+1}^{\pi_k})(s, a) - \sigma_{\mathcal{P}_h(s,a)}(V_{h+1}^{\pi_k})(s, a)) + b_h^k(s, a)\right].$$

By adding and subtracting a term $\sigma_{\mathcal{P}_h(s,a)}(\hat{V}_{h+1}^{\pi_k})(s, a)$, we have

$$\sigma_{\hat{\mathcal{P}}_h(s,a)}(\hat{V}_{h+1}^{\pi_k})(s, a) - \sigma_{\mathcal{P}_h(s,a)}(V_{h+1}^{\pi_k})(s, a)$$

$$= \sigma_{\hat{\mathcal{P}}_h(s,a)}(\hat{V}_{h+1}^{\pi_k})(s, a) - \sigma_{\mathcal{P}_h(s,a)}(\hat{V}_{h+1}^{\pi_k})(s, a) + \sigma_{\mathcal{P}_h(s,a)}(\hat{V}_{h+1}^{\pi_k})(s, a) - \sigma_{\mathcal{P}_h(s,a)}(V_{h+1}^{\pi_k})(s, a)$$

$$\leq \sigma_{\hat{\mathcal{P}}_h(s,a)}(\hat{V}_{h+1}^{\pi_k})(s, a) - \sigma_{\mathcal{P}_h(s,a)}(\hat{V}_{h+1}^{\pi_k})(s, a) + \max_{P_h \in \mathcal{P}_h} P_h(\cdot \mid s, a)(\hat{V}_{h+1}^{\pi_k} - V_{h+1}^{\pi_k}).$$

Let $p_h(\cdot \mid s, a) = \arg\max_{P_h \in \mathcal{P}_h} P_h(\cdot \mid s, a)(\hat{V}_{h+1}^{\pi_k} - V_{h+1}^{\pi_k})$, we have

$$(\hat{V}_h^{\pi_k} - V_h^{\pi_k})(s)$$

$$\leq \mathbb{E}_{\pi_k}\left[\hat{r}_h^k(s, a) - r_h^k(s, a) + \sigma_{\hat{\mathcal{P}}_h(s,a)}(\hat{V}_{h+1}^{\pi_k})(s, a) - \sigma_{\mathcal{P}_h(s,a)}(\hat{V}_{h+1}^{\pi_k})(s, a) + p_h(\cdot \mid s, a)(\hat{V}_{h+1}^{\pi_k} - V_{h+1}^{\pi_k}) + b_h^k(s, a)\right]$$

$$= \mathbb{E}_{\pi_k, p_h}\left[\hat{r}_h^k(s, a) - r_h^k(s, a) + \sigma_{\hat{\mathcal{P}}_h(s,a)}(\hat{V}_{h+1}^{\pi_k})(s, a) - \sigma_{\mathcal{P}_h(s,a)}(\hat{V}_{h+1}^{\pi_k})(s, a) + \hat{V}_{h+1}^{\pi_k}(s) - V_{h+1}^{\pi_k}(s) + b_h^k(s, a)\right]$$

By applying the above relation recursively and with the fact that for any policy $\pi$ and state $s$, $V_{H+1}^{\pi_k}(s) = \hat{V}_{H+1}^{\pi_k}(s) = 0$, we have

$$(\hat{V}_1^{\pi_k} - V_1^{\pi_k})(s) \leq \sum_{h=1}^{H} \mathbb{E}_{\pi_k, \{p_t\}_{t=1}^{h}}\left[\hat{r}_h^k(s, a) - r_h^k(s, a) + \sigma_{\hat{\mathcal{P}}_h(s,a)}(\hat{V}_{h+1}^{\pi_k})(s, a) - \sigma_{\mathcal{P}_h(s,a)}(\hat{V}_{h+1}^{\pi_k})(s, a) + b_h^k(s, a)\right].$$

Conditioned on the good event and by the design of our bonus function, we have

$$\hat{r}_h^k(s, a) - r_h^k(s, a) + \sigma_{\hat{\mathcal{P}}_h(s,a)}(\hat{V}_{h+1}^{\pi_k})(s, a) - \sigma_{\mathcal{P}_h(s,a)}(\hat{V}_{h+1}^{\pi_k})(s, a) \leq b_h^k(s, a).$$

Then, with probability at least $1 - \delta$, we have

$$\sum_{k=1}^{K}(\hat{V}_1^{\pi_k} - V_1^{\pi_k})(s) \leq \sum_{k=1}^{K}\sum_{h=1}^{H} \mathbb{E}_{\pi_k, \{p_t\}_{t=1}^{h}}\left[2b_h^k(s, a)\right]$$

$$\leq H\sqrt{K} + O\left(H\sqrt{S \log(SAH^2K^{3/2}(4 + \rho)/\delta)}\right) \sum_{k=1}^{K}\sum_{h=1}^{H} \mathbb{E}_{\pi_k, \{p_t\}_{t=1}^{h}}\left[\sqrt{\frac{1}{N_h^k(s, a)}}\right].$$

Let $\omega_h = \{p_t\}_{t=1}^{h}/P_h^o$, and assume $P_h^o(\cdot \mid s, a) \geq c, \forall(s, a)$. By Lemma E.2, we have the bound of the visitation counts:

$$\sum_{k=1}^{K}\sum_{h=1}^{H} \mathbb{E}_{\pi_k, \{p_t\}_{t=1}^{h}}\left[\sqrt{\frac{1}{N_h^k(s, a)}}\right] \leq \sum_{k=1}^{K}\sum_{h=1}^{H} \mathbb{E}_{\pi_k, P_h^o}\left[\omega_h\sqrt{\frac{1}{N_h^k(s, a)}}\right]$$

$$\leq \max_{h \in [H]} \omega_h \sum_{k=1}^{K}\sum_{h=1}^{H} \mathbb{E}_{\pi_k, P_h^o}\left[\sqrt{\frac{1}{N_h^k(s, a)}}\right]$$

$$\leq \frac{2H\sqrt{SAK}}{c}.$$

Combining everything, with probability at least $1 - \delta$

$$\sum_{k=1}^{K}(\hat{V}_1^{\pi_k} - V_1^{\pi_k})(s) = O\left(\frac{H^2S}{c}\sqrt{AK \log\left(SAH^2K^{3/2}(1 + \rho)/\delta\right)}\right).$$

$\square$

**Lemma A.4.** *For any $h, k, s, a$, the following inequality holds with probability at least $1 - \delta'$,*

$$\sigma_{\mathcal{P}_h(s,a)}(\hat{V}_{h+1}^{\pi_k})(s,a) - \sigma_{\hat{\mathcal{P}}_h(s,a)}(\hat{V}_{h+1}^{\pi_k})(s,a) \leq H\sqrt{\frac{4S\log(3SAH^3K^{3/2}(4+\rho)/\delta')}{N_h^k(s,a)}} + \frac{1}{H\sqrt{K}}\,.$$

*Proof.* By the definition of $\sigma_{\mathcal{P}_h(s,a)}(\hat{V}_{h+1}^{\pi_k})(s,a) = \min_{P_h \in \mathcal{P}_h}\sum_{s'} P_h(s' \mid s,a)\hat{V}_{h+1}^{\pi_k}(s')$, we have the following optimization problem:

$$\min_{P_h} \sum_{s'} P_h(s' \mid s,a)\hat{V}_{h+1}^{\pi_k}(s')$$

$$\text{s.t.} \quad \begin{cases} \sum_{s'}|P_h(s' \mid s,a) - P_h^o(s' \mid s,a)| \leq \rho\,, \\ \sum_{s'} P_h(s' \mid s,a) = 1\,, \\ P_h^o(\cdot \mid s,a) > 0, P_h(\cdot \mid s,a) \geq 0\,. \end{cases}$$

Define $\tilde{P}_h(s' \mid s,a) = \frac{P_h(s'|s,a)}{P_h^o(s'|s,a)}$, we can rewrite the above optimization problem as

$$\min_{\tilde{P}_h} \sum_{s'} \tilde{P}_h(s' \mid s,a)P_h^o(s' \mid s,a)\hat{V}_{h+1}^{\pi_k}(s')$$

$$\text{s.t.} \quad \begin{cases} \sum_{s'}|\tilde{P}_h(s' \mid s,a) - 1|P_h^o(s' \mid s,a) \leq \rho\,, \\ \sum_{s'} \tilde{P}_h(s' \mid s,a)P_h^o(s' \mid s,a) = 1\,, \\ \tilde{P}_h(s' \mid s,a) \geq 0 \quad \forall s' \in \mathcal{S}\,. \end{cases}$$

Using the Lagrangian multiplier method, we have the following Lagrangian $L(\tilde{P}_h, \eta, \lambda)$ with Lagrangian multiplier $\eta \in \mathbb{R}, \lambda \geq 0$,

$$L(\tilde{P}_h, \eta, \lambda)(s,a) = \sum_{s'} \tilde{P}_h(s' \mid s,a)P_h^o(s' \mid s,a)\hat{V}_{h+1}^{\pi_k}(s') + \lambda\left(\sum_{s'}|\tilde{P}_h(s' \mid s,a) - 1|P_h^o(s' \mid s,a) - \rho\right)$$

$$- \eta\left(\sum_{s'} \tilde{P}_h(s' \mid s,a)P_h^o(s' \mid s,a) - 1\right)$$

$$= \eta - \lambda\rho - \lambda\sum_{s'} P_h^o(s' \mid s,a)\left(\frac{\eta}{\lambda}\tilde{P}_h(s' \mid s,a) - |\tilde{P}_h(s' \mid s,a) - 1| - \frac{\tilde{P}_h(s' \mid s,a)\hat{V}_{h+1}^{\pi_k}(s')}{\lambda}\right)$$

$$= \eta - \lambda\rho - \lambda\sum_{s'} P_h^o(s' \mid s,a)\left(\frac{\eta - \hat{V}_{h+1}^{\pi_k}(s')}{\lambda}\tilde{P}_h(s' \mid s,a) - |\tilde{P}_h(s' \mid s,a) - 1|\right)\,.$$

We define $f(x) = |x-1|$ and the convex conjugate is $f^*(y) = \max_x\langle x, y\rangle - f(x)$. Let $x$ be $\tilde{P}_h$ and by using $f^*$, we can optimize over $\tilde{P}_h$ and rewrite the Lagrangian as

$$L(\eta, \lambda)(s,a) = \min_{\tilde{P}_h} L(\tilde{P}_h, \eta, \lambda)(s,a) = \eta - \lambda\rho - \lambda\sum_{s'} P_h^o(s' \mid s,a)f^*\left(\frac{\eta - \hat{V}_{h+1}^{\pi_k}(s')}{\lambda}\right)\,.$$

Notice that conditioned on $x \geq 0$, $f(x) = |x-1|$'s convex conjugate has the following closed form:

$$f^*(y) = \max_x\langle x, y\rangle - f(x) = \begin{cases} -1 & y \leq -1\,, \\ y & y \in [-1, 1]\,, \\ +\infty & y > 1\,. \end{cases}$$

Let $\tilde{\eta} = \eta + \lambda$, then using the closed form of $f^*(y)$, the equality $\max\{a, b\} = (a-b)_+ + b$ and condition on $\frac{\eta - \hat{V}_{h+1}^{\pi_k}(s')}{\lambda} \leq 1$,

we can rewrite the optimization problem as

$$L(\tilde{\eta}, \lambda)(s, a) = \eta - \lambda\rho - \lambda \sum_{s'} P_h^o(s' \mid s, a) f^* \left( \frac{\eta - \hat{V}_{h+1}^{\pi_k}(s')}{\lambda} \right)$$

$$= \tilde{\eta} - \lambda - \lambda\rho - \lambda \sum_{s'} P_h^o(s' \mid s, a) \max \left\{ \frac{\eta - \hat{V}_{h+1}^{\pi_k}(s')}{\lambda}, -1 \right\}$$

$$= \tilde{\eta} - \lambda - \lambda\rho - \lambda \sum_{s'} P_h^o(s' \mid s, a) \left( \left( \frac{\eta - \hat{V}_{h+1}^{\pi_k}(s')}{\lambda} - (-1) \right)_+ + (-1) \right)$$

$$= \tilde{\eta} - \lambda - \lambda\rho - \sum_{s'} P_h^o(s' \mid s, a)(\tilde{\eta} - \hat{V}_{h+1}^{\pi_k}(s'))_+ + \lambda$$

$$= \tilde{\eta} - \lambda\rho - \sum_{s'} P_h^o(s' \mid s, a)(\tilde{\eta} - \hat{V}_{h+1}^{\pi_k}(s'))_+ .$$

with the constraint of $\lambda$ being

$$\lambda \geq 0, \quad \tilde{\eta} - \min_s \hat{V}_{h+1}^{\pi_k}(s) \leq 2\lambda.$$

Then we discuss the constraint of $\tilde{\eta} = \eta + \lambda$ and show that $\tilde{\eta} \in R$. We discuss this by cases.

For any $x \leq \min_s \hat{V}_{h+1}^{\pi_k}(s)$, taking $\eta = x$, $\lambda = 0$, then we have $\tilde{\eta} = x$.

For any $x > \min_s \hat{V}_{h+1}^{\pi_k}(s)$, taking $\eta = \frac{x + \min_s \hat{V}_{h+1}^{\pi_k}(s)}{2}$, $\lambda = \frac{x - \min_s \hat{V}_{h+1}^{\pi_k}(s)}{2}$, then we have $\tilde{\eta} = x$.

Then we have $\tilde{\eta} \in R$. Fixing any $\tilde{\eta}$, from the definition of $L$, we need to choose $\lambda = \frac{(\tilde{\eta} - \min_s \hat{V}_{h+1}^{\pi_k}(s))_+}{2}$ to achieve the maximum of $L$. Then by directly optimizing it over $\lambda$, we can reduce the problem to

$$L(\tilde{\eta})(s, a) = \tilde{\eta} - \frac{(\tilde{\eta} - \min_s \hat{V}_{h+1}^{\pi_k}(s))_+}{2}\rho - \sum_{s'} P_h^o(s' \mid s, a)(\tilde{\eta} - \hat{V}_{h+1}^{\pi_k}(s'))_+ .$$

with the constraint $\tilde{\eta} \in R$.

Define the function $g$ as

$$g(\tilde{\eta}, P_h^o) = -L(\tilde{\eta})(s, a) = \sum_{s'} P_h^o(s' \mid s, a) \left( \tilde{\eta} - \hat{V}_{h+1}^{\pi_k}(s') \right)_+ - \tilde{\eta} + \frac{(\tilde{\eta} - \min_s \hat{V}_{h+1}^{\pi_k}(s))_+}{2}\rho .$$

Then we investigate the optimum of $g$. First notice that $g(0) = 0$, when $\tilde{\eta} \leq 0$, $g(\tilde{\eta}, P_h^o) = -\tilde{\eta} \geq 0$.

On the other hand, when $\tilde{\eta} \geq H$,

$$g(\tilde{\eta}, P_h^o) = \sum_{s'} P_h^o(s' \mid s, a)(\tilde{\eta} - \hat{V}_{h+1}^{\pi_k}(s')) - \tilde{\eta} + \frac{(\tilde{\eta} - \min_s \hat{V}_{h+1}^{\pi_k}(s))}{2}\rho$$

$$= -\sum_{s'} P_h^o(s' \mid s, a)\hat{V}_{h+1}^{\pi_k}(s') + \frac{(\tilde{\eta} - \min_s \hat{V}_{h+1}^{\pi_k}(s))}{2}\rho .$$

Note that now $g$ is directly proportional to $\tilde{\eta}$, therefore $g$ achieves the minimum within the range of $\tilde{\eta} \in [0, H]$. We remark that the same form is also used for analyzing robust policy evaluation (Lemma B.1 [Yang et al., 2021]).

With this, we can rewrite

$$\sigma_{\hat{\mathcal{P}}_h(s,a)}(\hat{V}_{h+1}^{\pi_k})(s, a) - \sigma_{\mathcal{P}_h(s,a)}(\hat{V}_{h+1}^{\pi_k})(s, a) = -\min_{\eta_1 \in [0, H]} g(\eta_1, \hat{P}_h^{o,k}) + \min_{\eta_2 \in [0, H]} g(\eta_2, P_h^o)$$

$$\leq \max_{\eta \in [0, H]} |g(\eta, \hat{P}_h^{o,k}) - g(\eta, P_h^o)| .$$

To upper bound $\sigma_{\hat{\mathcal{P}}_h(s,a)}(\hat{V}_{h+1}^{\pi_k})(s,a) - \sigma_{\mathcal{P}_h(s,a)}(\hat{V}_{h+1}^{\pi_k})(s,a)$, we first upper bound $|g\left(\eta, \hat{P}_h^{o,k}\right) - g\left(\eta, P_h^o\right)|$.

$$
\begin{aligned}
|g\left(\eta, \hat{P}_h^{o,k}\right) - g\left(\eta, P_h^o\right)| &= \left| \sum_{s'} \hat{P}_h^{o,k}(s' \mid s,a)\left(\eta - \hat{V}_{h+1}^{\pi_k}(s')\right)_+ - \sum_{s'} P_h^o(s' \mid s,a)\left(\eta - \hat{V}_{h+1}^{\pi_k}(s')\right)_+ \right| \\
&\leq \left\| \hat{P}_h^{o,k}(\cdot \mid s,a) - P_h^o(\cdot \mid s,a) \right\|_1 \max_{s \in \mathcal{S}} |\eta - \hat{V}_{h+1}^{\pi_k}(s)|_\infty \\
&\leq H \left\| \hat{P}_h^{o,k}(\cdot \mid s,a) - P_h^o(\cdot \mid s,a) \right\|_1,
\end{aligned}
$$

where the first inequality is by Cauchy-Schwarz inequality, the second inequality follows from $\eta \in [0, H]$.

By Hoeffding's inequality and an union bound over all $s, a$, the following inequality holds with probability at least $1 - \delta'$:

$$
\left\| \hat{P}_h^{o,k}(\cdot \mid s,a) - P_h^k(\cdot \mid s,a) \right\|_1 \leq \sqrt{\frac{4S \log(3SAH^2K/\delta')}{N_h^k(s,a)}}.
$$

To upper bound the error with maximum over $\eta$, we first create an $\epsilon$-net $N_\epsilon(\eta)$ with $g$ over $\eta \in [0, H]$ such that

$$
\max_{\eta \in [0,H]} |g\left(\eta, \hat{P}_h^{o,k}\right) - g\left(\eta, P_h^o\right)| \leq \max_{\eta \in N_\epsilon(\eta)} |g\left(\eta, \hat{P}_h^{o,k}\right) - g\left(\eta, P_h^o\right)| + 2\epsilon.
$$

By taking an union bound over $N_\epsilon(\eta)$, we have

$$
\max_{\eta \in [0,H]} |g\left(\eta, \hat{P}_h^{o,k}\right) - g\left(\eta, P_h^o\right)| \leq H \sqrt{\frac{4S \log(3SAH^2K|N_\epsilon(\eta)|/\delta')}{N_h^k(s,a)}} + 2\epsilon,
$$

where $|N_\epsilon(\eta)|$ is the size of the $\epsilon$-net.

It now remains to bound the size of $|N_\epsilon(\eta)|$, which can be obtained easily if $g$ is Lischitz. Notice that

$$
\begin{aligned}
|g(\tilde{\eta}_1, P_h^o) - g(\tilde{\eta}_2, P_h^o)| &\leq \sum_{s'} P_h^o(s' \mid s,a)|\tilde{\eta}_1 - \tilde{\eta}_2| + |\tilde{\eta}_1 - \tilde{\eta}_2| + \frac{|\tilde{\eta}_1 - \tilde{\eta}_2|}{2}\rho \\
&= \frac{4+\rho}{2}|\tilde{\eta}_1 - \tilde{\eta}_2|,
\end{aligned}
$$

where the first inequality is by the absolute inequality and $|(a)_+ - (b)_+| \leq |a - b|$.

Then $g$ is a $\frac{4+\rho}{2}$-Lipschitz function over $\eta \in [0, H]$, thus combined with Lemma E.1, we have $|N_\epsilon(\eta)| = O\left(\frac{4+\rho}{2\epsilon}\right)$. Hence, we have the following inequality happens with at least $1 - \delta'$ probability:

$$
\max_{\eta \in [0,H]} |g\left(\eta, \hat{P}_h^{o,k}\right) - g\left(\eta, P_h^o\right)| \leq H \sqrt{\frac{4S \log(3SAH^2K(4+\rho)/2\epsilon\delta')}{N_h^k(s,a)}} + 2\epsilon.
$$

Take $\epsilon = \frac{1}{2\sqrt{K}}$, we have the following inequality happens with at least $1 - \delta'$ probability:

$$
\begin{aligned}
\sigma_{\mathcal{P}_h(s,a)}(\hat{V}_{h+1}^{\pi_k})(s,a) - \sigma_{\hat{\mathcal{P}}_h(s,a)}(\hat{V}_{h+1}^{\pi_k})(s,a) &\leq \max_{\eta \in [0,H]} |g\left(\eta, \hat{P}_h^{o,k}\right) - g\left(\eta, P_h^o\right)| \\
&\leq H \sqrt{\frac{4S \log(3SAH^2K^{3/2}(4+\rho)/\delta')}{N_h^k(s,a)}} + \frac{1}{\sqrt{K}}.
\end{aligned}
$$

$\square$

# B  PROOF OF THEOREM 5.2

## B.1  GOOD EVENTS

We first define the following good events, in which case we estimate the reward function and the nominal transition functions fairly accurately.

$$\mathcal{G}_k^r = \left\{ \forall s, a, h : \left| r_h(s,a) - \hat{r}_h^k(s,a) \right| \leq \sqrt{\frac{2\ln(2SAH^2K/\delta')}{N_h^k(s,a)}} \right\},$$

$$\mathcal{G}_k^p = \left\{ \forall s, a, h : \sigma_{\mathcal{P}_h(s)}(\hat{V}_{h+1}^{\pi_k})(s,a) - \sigma_{\hat{\mathcal{P}}_h(s)}(\hat{V}_{h+1}^{\pi_k})(s,a) \leq C_h^k(s,a) \right\},$$

where

$$C_h^k(s,a) = AH\sqrt{\frac{4SA\log(3SA^2H^3K^{3/2}(4+\rho)/\delta')}{N_h^k(s,a)}} + \frac{1}{H\sqrt{K}}.$$

When the two good events happens at the same time, we say the algorithm in inside the good event $\mathcal{G} = \left( \bigcap_{k=1}^K \mathcal{G}_k^r \right) \cap \left( \bigcap_{k=1}^K \mathcal{G}_k^p \right)$. The following lemma shows that $\mathcal{G}$ happens with high probability.

**Lemma B.1** (Good event). *Let $\delta = 2\delta'$, then the good event happens with high probability, i.e. $\mathbb{P}\left[\mathcal{G}\right] \geq 1 - \delta$.*

*Proof.* By Hoeffding's inequality and an union bound on all $s, a$, all possible values of $N_k(s,a)$ and $k$, we have $\mathbb{P}\left[\bigcap_{k=1}^K \mathcal{G}_k^r\right] \geq 1 - \delta'$. By Lemma B.3, we have $\mathbb{P}\left[\bigcap_{k=1}^K \mathcal{G}_k^p\right] \geq 1 - \delta'$ Then set $\delta = 2\delta'$ and we have the desired result. □

## B.2  DESIGN OF THE BONUS FUNCTION

In the case of $s$-rectangular uncertainty set, we use the following bonus function $b_h^k(s,a)$ to encourage exploration.

$$b_h^k(s,a) = AH\sqrt{\frac{4SA\log(3SA^2H^2K^{3/2}(4+\rho)/\delta)}{N_h^k(s,a)}} + \frac{1}{\sqrt{K}} + \sqrt{\frac{2\log(3SAH^2K/\delta')}{N_h^k(s,a)}}. \tag{6}$$

## B.3  REGRET ANALYSIS

*Proof.* Similar to the case of $(s,a)$-rectangular set, we start with decomposing the regret as follows,

$$\text{Regret}(K) = \sum_{k=1}^K V_1^*(s) - V_1^{\pi_k}(s)$$

$$= \sum_{k=1}^K \left( V_1^*(s) - \hat{V}_1^{\pi_k}(s) \right) + \left( \hat{V}_1^{\pi_k}(s) - V_1^{\pi_k}(s) \right).$$

By Lemma A.2 and Lemma B.2, with probability at least $1 - \delta$, we have

$$\text{Regret}(K) = O\left( \frac{H^2\sqrt{K\log A}}{c} \right) + O\left( \frac{SA^2H^2}{c}\sqrt{K\log(SA^2H^2K^{3/2}(1+\rho)/\delta)} \right)$$

$$= O\left( \frac{SA^2H^2}{c}\sqrt{K\log(SA^2H^2K^{3/2}(1+\rho)/\delta)} \right).$$

□

**Lemma B.2.** *With Algorithm 1, we have*

$$\sum_{k=1}^K (\hat{V}_1^{\pi_k} - V_1^{\pi_k})(s) = O\left( \frac{SA^2H^2}{c}\sqrt{K\log(SA^2H^2K^{3/2}(1+\rho)/\delta)} \right).$$

*Proof.* Similar to the case with $(s,a)$-rectangular uncertainty set, for any $k$, we can decompose $(\hat{V}_1^{\pi_k} - \hat{V}_1^{\pi_k})(s)$ as,

$$(\hat{V}_1^{\pi_k} - \hat{V}_1^{\pi_k})(s)$$

$$\leq \sum_{h=1}^{H} \mathbb{E}_{\pi_k,\{p_t\}_{t=1}^{h}} \left[ (r_h^k(s,a) - \hat{r}_h^k(s,a)) + \left( \sigma_{\hat{\mathcal{P}}_h(s)}\left(\hat{V}_{h+1}^{\pi_k}\right)(s,a) - \sigma_{\mathcal{P}_h(s)}\left(\hat{V}_{h+1}^{\pi_k}\right)(s,a) \right) + b_h^k(s,a) \right].$$

Thus by the design of our bonus function and with probability at least $1 - \delta$, we have

$$\sum_{k=1}^{K}(\hat{V}_1^{\pi_k} - V_1^{\pi_k})(s)$$

$$\leq 2 \sum_{k=1}^{K} \sum_{h=1}^{H} \mathbb{E}_{\pi_k,\{p_t\}_{t=1}^{h}} \left[ b_h^k(s,a) \right]$$

$$= H\sqrt{K} + O\left( HA\sqrt{SA\log(SA^2H^2K^{3/2}(1+\rho)/\delta)} \right) \sum_{k=1}^{K} \sum_{h=1}^{H} \mathbb{E}_{\pi_k,\{p_t\}_{t=1}^{h}} \left[ \sqrt{\frac{1}{N_h^k(s,a)}} \right].$$

Let $\omega_h = \{p_t\}_{t=1}^{h}/P_h^o$, and assume $P_h^o(\cdot \mid s,a) \geq c, \forall(s,a)$. By Lemma E.2, we have the bound of the visitation counts:

$$\sum_{k=1}^{K} \sum_{h=1}^{H} \mathbb{E}_{\pi_k,\{p_t\}_{t=1}^{h}} \left[ \sqrt{\frac{1}{N_h^k(s,a)}} \right] \leq \sum_{k=1}^{K} \sum_{h=1}^{H} \mathbb{E}_{\pi_k,P_h^o} \left[ \omega_h \sqrt{\frac{1}{N_h^k(s,a)}} \right]$$

$$\leq \max_{h\in[H]} \omega_h \sum_{k=1}^{K} \sum_{h=1}^{H} \mathbb{E}_{\pi_k,P_h^o} \left[ \sqrt{\frac{1}{N_h^k(s,a)}} \right]$$

$$\leq \frac{2H\sqrt{SAK}}{c}.$$

Combining everything, conditioned on the good event we have

$$\sum_{k=1}^{K}(\hat{V}_1^{\pi_k} - V_1^{\pi_k})(s) = O\left( \frac{SA^2H^2}{c} \sqrt{K\log(SA^2H^2K^{3/2}(1+\rho)/\delta)} \right).$$

$\square$

**Lemma B.3.** *For any $h, k, s, a$, the following inequality holds with probability at least $1 - \delta$,*

$$\sigma_{\hat{\mathcal{P}}_h(s)}(\hat{V}_{h+1}^{\pi_k})(s,a) - \sigma_{\mathcal{P}_h(s)}(\hat{V}_{h+1}^{\pi_k})(s,a) \leq AH\sqrt{\frac{4SA\log(3SA^2H^2K^{3/2}(4+\rho)/\delta)}{N_h^k(s,a)}} + \frac{1}{\sqrt{K}}.$$

*Proof.* By the definition of $\sigma_{\mathcal{P}_h(s)}(\hat{V}_{h+1}^{\pi_k})(s,a) = \inf_{P_h \in \mathcal{P}_h} \sum_{s'} P_h(s' \mid s,a)\hat{V}_{h+1}^{\pi_k}(s')$, we consider the following optimization problem:

$$\min_{P_h} \sum_{s'} P_h(s' \mid s,a)\hat{V}_{h+1}^{\pi_k}(s')$$

$$\text{s.t.} \quad \begin{cases} \sum_{s',a'} |P_h(s' \mid s,a') - P_h^o(s' \mid s,a')| \leq A\rho, \\ \sum_{s'} P_h(s' \mid s,a') = 1, \forall a' \in \mathcal{A}, \\ P_h^o(\cdot \mid s,a') > 0, P_h(\cdot \mid s,a') \geq 0, \forall a' \in \mathcal{A}. \end{cases}$$

Let $\tilde{P}_h(s' \mid s,a) = \frac{P_h(s' \mid s,a)}{P_h^o(s' \mid s,a)}$, we can rewrite the above optimization problem as

$$\min_{\tilde{P}_h} \sum_{s'} \tilde{P}_h(s' \mid s,a)P_h^o(s' \mid s,a)\hat{V}_{h+1}^{\pi_k}(s')$$

$$\text{s.t.} \quad \begin{cases} \sum_{s',a'} |(\tilde{P}_h(s' \mid s,a') - 1|P_h^o(s' \mid s,a') \leq A\rho, \\ \sum_{s'} \tilde{P}_h(s' \mid s,a')P_h^o(s' \mid s,a') = 1, \quad \forall a' \in \mathcal{A} \\ \tilde{P}_h(\cdot \mid s,a') \geq 0, \quad \forall a' \in \mathcal{A}. \end{cases}$$

Use the Lagrangian multiplier method and $f(x) = |x - 1|$, we have the Lagrangian $L(\tilde{P}_h, \eta, \lambda)$ with multiplier $\eta = \{\eta_a\}_{a \in \mathcal{A}}, \eta_a \in \mathbb{R}, \lambda \geq 0$,

$$L\left(\tilde{P}_h, \eta, \lambda\right)(s, a)$$

$$= \sum_{s'} \tilde{P}_h(s' \mid s, a) P_h^o(s' \mid s, a) \hat{V}_{h+1}^{\pi_k}(s') + \lambda \left( \sum_{s', a'} \left|(\tilde{P}_h(s' \mid s, a') - 1\right| P_h^o(s' \mid s, a') - A\rho \right)$$

$$- \sum_{a'} \eta_{a'} \left( \sum_{s'} \tilde{P}_h(s' \mid s, a') P_h^o(s' \mid s, a') - 1 \right)$$

$$= -\lambda A\rho + \sum_{a'} \eta_{a'} + \lambda \sum_{s', a'} P_h^o(s' \mid s, a') \left( f\left(\tilde{P}_h(s' \mid s, a')\right) - \tilde{P}_h(s' \mid s, a') \left( \frac{\eta_{a'} - \mathbb{I}\{a' = a\} V_{h+1}^{\pi_k}(s')}{\lambda} \right) \right).$$

The convex conjugate of $f$ is $f^*(y) = \max_x \langle x, y \rangle - f(x)$. Using $f^*$, we can thus optimize over $\tilde{P}_h$ and rewrite the Lagrangian over as

$$L(\eta, \lambda)(s, a) = \min_{\tilde{P}_h} L\left(\tilde{P}_h, \eta, \lambda\right)(s, a)$$

$$= -\lambda A\rho + \sum_{a'} \eta_{a'} - \lambda \sum_{s', a'} P_h^o(s' \mid s, a') f^* \left( \frac{\eta_{a'} - \mathbb{I}\{a' = a\} V_{h+1}^{\pi_k}(s')}{\lambda} \right).$$

Conditioned on $x \geq 0$, $f(x) = |x - 1|$, notice that the conjugate $f^*(y)$ has the following closed form,

$$f^*(y) = \max_x \langle x, y \rangle - f(x) = \begin{cases} -1 & y \leq -1, \\ y & y \in [-1, 1], \\ +\infty & y > 1. \end{cases}$$

Let $\tilde{\eta}_a = \eta_a + \lambda$, using the closed form of $f^*(y)$, the equality $\max\{a, b\} = (a - b)_+ + b$ and conditioned on $\frac{\eta_{a'} - \mathbb{I}\{a' = a\} V_{h+1}^{\pi_k}(s')}{\lambda} \leq 1$, we can rewrite the optimization problem as

$$L(\tilde{\eta}, \lambda)(s, a) = -\lambda A\rho + \sum_{a'} \eta_{a'} - \lambda \sum_{s', a'} P_h^o(s' \mid s, a') f^* \left( \frac{\eta_{a'} - \mathbb{I}\{a' = a\} V_{h+1}^{\pi_k}(s')}{\lambda} \right)$$

$$= -\lambda A\rho - \lambda A + \sum_{a'} \tilde{\eta}_{a'} - \lambda \sum_{s', a'} P_h^o(s' \mid s, a') \max \left\{ \frac{\eta_{a'} - \mathbb{I}\{a' = a\} V_{h+1}^{\pi_k}(s')}{\lambda}, -1 \right\}$$

$$= -\lambda A\rho + \sum_{a'} \tilde{\eta}_{a'} - \sum_{s', a'} P_h^o(s' \mid s, a') \left( \tilde{\eta}_{a'} - \mathbb{I}\{a' = a\} V_{h+1}^{\pi_k}(s') \right)_+.$$

where constraint of $\lambda$ is

$$\lambda \geq 0, \quad \tilde{\eta}_{a'} - \mathbb{I}\{a' = a\} V_{h+1}^{\pi_k}(s') \leq 2\lambda, \ \forall a', s'.$$

Note that the above Lagrangian is inversely proportional to $\lambda$ and it achieves the maximum when $\lambda = \max_{s', a'} \frac{(\tilde{\eta}_{a'} - \mathbb{I}\{a' = a\} V_{h+1}^{\pi_k}(s'))_+}{2}$. Directly optimize over $\lambda$, we can reduce the problem to

$$L(\tilde{\eta})(s, a) = \sum_{a'} \tilde{\eta}_{a'} - \sum_{s', a'} P_h^o(s' \mid s, a') \left( \tilde{\eta}_{a'} - \mathbb{I}\{a' = a\} V_{h+1}^{\pi_k}(s') \right)_+ - \max_{s', a'} \frac{A\rho(\tilde{\eta}_{a'} - \mathbb{I}\{a' = a\} V_{h+1}^{\pi_k}(s'))_+}{2}.$$

Define $g\left(\tilde{\eta}, P_h^o\right) = -L(\tilde{\eta})(s, a)$ as

$$g(\tilde{\eta}, P_h^o) = -\sum_{a'} \tilde{\eta}_{a'} + \sum_{s', a'} P_h^o(s' \mid s, a') \left( \tilde{\eta}_{a'} - \mathbb{I}\{a' = a\} V_{h+1}^{\pi_k}(s') \right)_+ + \max_{s', a'} \frac{A\rho(\tilde{\eta}_{a'} - \mathbb{I}\{a' = a\} V_{h+1}^{\pi_k}(s'))_+}{2}.$$

Assume $g$ achieves its minimum when $\tilde{\eta} = \{\tilde{\eta}_1, \cdots, \tilde{\eta}_A\}$. Suppose $\tilde{\eta}$ has a component $\tilde{\eta}_a < 0$. Consider $\eta' = \{\tilde{\eta}_1, \cdots, 0, \cdots, \tilde{\eta}_a\}$, where we change the zero element $\tilde{\eta}_a$ to 0 and keep other components unchanged. Then we have

$$g(\tilde{\eta}, P_h^o) - g(\eta', P_h^o) = -\tilde{\eta}_A > 0\,,$$

which contradict with the hypothesis that $g$ achieves its minimum in $\tilde{\eta}$.

On the other hand, suppose $\tilde{\eta}$ has a component $\tilde{\eta}_a > H$. Then consider $\eta' = \{\tilde{\eta}_1, \cdots, H, \cdots, \tilde{\eta}_a\}$, where we change corresponding $\tilde{\eta}_a$ to 0 and keep other components unchanged. Denote $f(\tilde{\eta}) = \max_{s',a'} \frac{A\rho(\tilde{\eta}_{a'} - \mathbb{I}\{a'=a\}V_{h+1}^{\pi_k}(s'))_+}{2}$, and we have

$$
\begin{aligned}
g\left(\tilde{\eta}, P_h^o\right) - g\left(\eta', P_h^o\right) &= -\tilde{\eta}_A + H + \sum_{s'} P_h^o(s' \mid s,a)(\tilde{\eta}_a - H) + f(\tilde{\eta}) - f(\eta') \\
&\geq -\tilde{\eta}_A + H + \sum_{s'} P_h^o(s' \mid s,a)(\tilde{\eta}_a - H) \\
&= 0\,.
\end{aligned}
$$

Therefore, $g$ achieves its minimum with $\tilde{\eta}$, with $0 \leq \eta_a \leq H, \forall a \in \mathcal{A}$. We remark that a similar form and technique are also used for analyzing robust policy evaluation (Lemma C.1 [Yang et al., 2021]).

We can now rewrite

$$
\begin{aligned}
\sigma_{\hat{\mathcal{P}}_h(s)}\left(\hat{V}_{h+1}^{\pi_k}\right)(s,a) - \sigma_{\mathcal{P}_h(s)}\left(\hat{V}_{h+1}^{\pi_k}\right)(s,a) &= \min_{\eta_1 \in [0,H]^{|\mathcal{A}|}} g(\eta_1, \hat{P}_h^{o,k}) - \min_{\eta_2 \in [0,H]^{|\mathcal{A}|}} g(\eta_2, P_h^o) \\
&\leq \max_{\eta \in [0,H]^{|\mathcal{A}|}} \left| g\left(\eta, \hat{P}_h^{o,k}\right) - g\left(\eta, P_h^o\right) \right|\,.
\end{aligned}
$$

To upper bound $\sigma_{\hat{\mathcal{P}}_h(s)}\left(\hat{V}_{h+1}^{\pi_k}\right)(s,a) - \sigma_{\mathcal{P}_h(s)}\left(\hat{V}_{h+1}^{\pi_k}\right)(s,a)$, we first consider the bound of $\left| g\left(\eta, \hat{P}_h^{o,k}\right) - g\left(\eta, P_h^o\right) \right|$,

$$
\begin{aligned}
&\left| g\left(\eta, \hat{P}_h^{o,k}\right) - g\left(\eta, P_h^o\right) \right| \\
&= \left| \sum_{s',a'} \hat{P}_h^{o,k}(s' \mid s,a')\left(\eta_{a'} - \mathbb{I}\{a'=a\}V_{h+1}^{\pi_k}(s')\right)_+ - \sum_{s',a'} P_h^o(s' \mid s,a')\left(\eta_{a'} - \mathbb{I}\{a'=a\}V_{h+1}^{\pi_k}(s')\right)_+ \right| \\
&= \left| \sum_{a'} \sum_{s'} \left(\hat{P}_h^{o,k}(s' \mid s,a') - P_h^o(s' \mid s,a')\right)\left(\eta_{a'} - \mathbb{I}\{a'=a\}V_{h+1}^{\pi_k}(s')\right)_+ \right| \\
&\leq \sum_{a'} \left\| \hat{P}_h^{o,k}(\cdot \mid s,a') - P_h^o(\cdot \mid s,a') \right\|_1 \max_{s \in \mathcal{S}} \left| \eta_{a'} - \mathbb{I}\{a'=a\}V_{h+1}^{\pi_k}(s) \right| \\
&\leq H \sum_{a'} \left\| \hat{P}_h^{o,k}(\cdot \mid s,a') - P_h^o(\cdot \mid s,a') \right\|_1\,,
\end{aligned}
$$

where the first inequality is by Cauchy-Schwarz inequality, the second inequality follows from $\eta_a \in [0,H], \forall a \in \mathcal{A}$.

By Hoeffding's inequality and an union bound over all $s, a', N_h^k(s,a)$, the following inequality holds with probability at least $1 - \delta$,

$$\left\| \hat{P}_h^{o,k}(\cdot \mid s,a') - P_h^o(\cdot \mid s,a') \right\|_1 \leq \sqrt{\frac{4S\log(SAH^2K/\delta)}{N_h^k(s,a)}}\,.$$

To upper bound $\max_{\eta \in [0,H]^{|\mathcal{A}|}} \left| g\left(\eta, \hat{P}_h^{o,k}\right) - g\left(\eta, P_h^o\right) \right|$, we first create an $\epsilon$-net $N_\epsilon(\eta)$ with $g$ over $\eta \in [0,H]$ such that

$$\max_{\eta \in [0,H]} \left| g\left(\eta, \hat{P}_h^{o,k}\right) - g\left(\eta, P_h^o\right) \right| \leq \max_{\eta \in N_\epsilon(\eta)} \left| g\left(\eta, \hat{P}_h^{o,k}\right) - g\left(\eta, P_h^o\right) \right| + 2\epsilon\,.$$

Taking an union bound over $N_\epsilon(\eta)$, we have

$$\max_{\eta \in [0,H]} \left| g\left(\eta, \hat{P}_h^{o,k}\right) - g\left(\eta, P_h^o\right) \right| \le HA \sqrt{\frac{4S \log(3SAH^2K|N_\epsilon(\eta)|/\delta)}{N_h^k(s,a)}} + 2\epsilon\,,$$

where $|N_\epsilon(\eta)|$ is the size of the $\epsilon$-net.

It now remains to find the size of the $\epsilon$-net, which can be easily obtained if $g$ is Lipschitz. Notice that

$$|g(\tilde{\eta}_1, P_h^o) - g(\tilde{\eta}_2, P_h^o)|$$

$$\le \sum_{s',a'} P_h^o(s' \mid s, a)|\tilde{\eta}_{1,a'} - \tilde{\eta}_{2,a'}| + \sum_{a'} |\tilde{\eta}_{1,a'} - \tilde{\eta}_{2,a'}| + \frac{\max_{a'} |\tilde{\eta}_{1,a'} - \tilde{\eta}_{2,a'}|}{2} A\rho$$

$$\le \frac{A(4 + \rho)}{2} \|\tilde{\eta}_1 - \tilde{\eta}_2\|_\infty\,,$$

where the first inequality is by the absolute inequality, the property of maximum function and $|(a)_+ - (b)_+| \le |a - b|$, the second inequality follows from the definition of infinity norm.

Therefore $g$ is a $\frac{A(4+\rho)}{2}$-Lipschitz function over $\eta \in [0, H]$. Thus combining with Lemma E.1, we have $|N_\epsilon(\eta)| \le \left(\frac{A(4+\rho)}{2\epsilon}\right)^A$. Hence, we have the following inequality happens with at least $1 - \delta'$ probability:

$$\sigma_{\hat{\mathcal{P}}_h(s)}(\hat{V}_{h+1}^{\pi_k})(s,a) - \sigma_{\mathcal{P}_h(s)}(\hat{V}_{h+1}^{\pi_k})(s,a) \le \max_{\eta_a \in [0,H]^{|\mathcal{A}|}} \left| g\left(\eta, \hat{P}_h^{o,k}\right) - g\left(\eta, P_h^o\right) \right|$$

$$\le AH \sqrt{\frac{4SA \log(3SA^2H^2K(4+\rho)/2\epsilon\delta')}{N_h^k(s,a)}} + 2\epsilon\,.$$

Take $\epsilon = \frac{1}{2\sqrt{K}}$, then

$$\sigma_{\hat{\mathcal{P}}_h(s)}(\hat{V}_{h+1}^{\pi_k})(s,a) - \sigma_{\mathcal{P}_h(s)}(\hat{V}_{h+1}^{\pi_k})(s,a) \le AH \sqrt{\frac{4SA \log(3SA^2H^2K^{3/2}(4+\rho)/\delta')}{N_h^k(s,a)}} + \frac{1}{\sqrt{K}}\,.$$

$\square$

# C EXTENSION TO UNCERTAINTY SET WITH KL DIVERGENCE

In this section, we extend our algorithm and analysis to uncertainty sets with KL divergence as a distance metric. We first formally define the uncertainty set considered, which is similar to the one in Definition 3.1.

**Definition C.1** ($(s,a)$-rectangular uncertainty set Iyengar [2005], Wiesemann et al. [2013]). *For all time step $h$ and with a given state-action pair $(s,a)$, the $(s,a)$-rectangular uncertainty set $\mathcal{P}_h(s,a)$ is defined as*

$$\mathcal{P}_h(s,a) = \{D_{KL}\left(P_h(\cdot \mid s,a), P_h^o(\cdot \mid s,a)\right) \leq \rho, P_h(\cdot \mid s,a) \in \Delta(\mathcal{S})\} ,$$

*where $P_h^o$ is the nominal transition kernel at $h$, $P_h^o(\cdot \mid s,a) > 0, \forall (s,a) \in \mathcal{S} \times \mathcal{A}$, $\rho$ is the level of uncertainty and $D_{KL}\left(p(\cdot \mid s,a), q(\cdot \mid s,a)\right) = \sum_{s' \in \mathcal{S}} p(s' \mid s,a) \log\left(\frac{p(s'\mid s,a)}{q(s'\mid s,a)}\right)$.*

With the above described uncertainty set, our algorithm solves $\sigma_{\hat{\mathcal{P}}_h}(\hat{V}_{h+1}^\pi)(s,a)$ by solving the following sub-problem,

$$\min_\lambda \lambda\rho + \lambda \log\left(\sum_{s'} \hat{P}_h^o(s' \mid s,a) \exp\left(\frac{-\hat{V}_{h+1}^{\pi_k}(s')}{\lambda}\right)\right) .$$

Our algorithm also uses the following bonus function in the robust policy evaluation step,

$$b_h^k(s,a) = C_h^k(s,a) + \sqrt{\frac{2\log(3SAH^2K/\delta')}{N_h^k(s,a)}} .$$

With these modifications to algorithm 1, the following theorem states the formal regret guarantee.

**Theorem C.2** (Regret under KL divergence $(s,a)$-rectangular uncertainty set). *Setting the learning rate $\beta = \sqrt{\frac{2\log A}{H^2 K}}$, then with probability at least $1 - \delta$, the regret incurred by Algorithm over $K$ episodes is bounded by*

$$Regret(K) = O\left(\frac{SH}{\rho c^2}\sqrt{AK\log(SAH^4K^{3/2}/\delta)}\right) ,$$

*where $0 < c \leq 1$ the minimal element of $P_h^o$, over all $h \in [H]$.*

In the following, we present the detailed analysis of Theorem C.2

## C.1 GOOD EVENTS

We first define the following good events, in which case we estimate the reward function and the nominal transition functions fairly accurately.

$$\mathcal{G}_k^r = \left\{\forall s,a,h : \left|r_h(s,a) - \hat{r}_h^k(s,a)\right| \leq \sqrt{\frac{2\ln(2SAH^2K/\delta')}{N_h^k(s,a)}}\right\} ,$$

$$\mathcal{G}_k^p = \left\{\forall s,a,h : \sigma_{\mathcal{P}_h(s)}(\hat{V}_{h+1}^{\pi_k})(s,a) - \sigma_{\hat{\mathcal{P}}_h(s)}(\hat{V}_{h+1}^{\pi_k})(s,a) \leq C_h^k(s,a)\right\} ,$$

where

$$C_h^k(s,a) = \frac{2H}{\rho c}\sqrt{\frac{4S\log(8SAH^4K^2/\delta'\rho)}{N_h^k(s,a)}} + \frac{1}{\sqrt{K}} ,$$

and $c$ is the minimal element of $P_h^o$, over all $h \in [H]$. When the two good events happens at the same time, we say the algorithm in inside the good event $\mathcal{G} = \left(\bigcap_{k=1}^K \mathcal{G}_k^r\right) \bigcap \left(\bigcap_{k=1}^K \mathcal{G}_k^p\right)$. The following lemma shows that $\mathcal{G}$ happens with high probability.

**Lemma C.3** (Good event). *Let $\delta = 2\delta'$, then the good event happens with high probability, i.e. $\mathbb{P}\left[\mathcal{G}\right] \geq 1 - \delta$.*

*Proof.* By Hoeffding's inequality and an union bound on all $s,a$, all possible values of $N_k(s,a)$ and $k$, we have $\mathbb{P}\left[\bigcap_{k=1}^K \mathcal{G}_k^r\right] \geq 1 - \delta'$. By Lemma C.5, we have $\mathbb{P}\left[\bigcap_{k=1}^K \mathcal{G}_k^p\right] \geq 1 - \delta'$ Then set $\delta = 2\delta'$ and we have the desired result. □

## C.2 REGRET ANALYSIS

*Proof.* Similar to the case of $(s, a)$-rectangular set, we start with decomposing the regret as follows,

$$\text{Regret}(K) = \sum_{k=1}^{K} V_1^*(s) - V_1^{\pi_k}(s)$$

$$= \sum_{k=1}^{K} \left( V_1^*(s) - \hat{V}_1^{\pi_k}(s) \right) + \left( \hat{V}_1^{\pi_k}(s) - V_1^{\pi_k}(s) \right) .$$

By Lemma A.2 and Lemma C.4, with probability at least $1 - \delta$, we have

$$\text{Regret}(K) = O \left( \frac{H^2 \sqrt{K \log A}}{c} \right) + O \left( \frac{SH}{\rho c^2} \sqrt{AK \log(SAH^4 K^{3/2}/\delta)} \right)$$

$$= O \left( \frac{SH}{\rho c^2} \sqrt{AK \log(SAH^4 K^{3/2}/\delta)} \right) ,$$

where $c$ is the minimal element of $P_h^o$, over all $h \in [H]$. $\qquad\square$

**Lemma C.4.** *With Algorithm 1, we have*

$$\sum_{k=1}^{K} (\hat{V}_1^{\pi_k} - V_1^{\pi_k})(s) = O \left( \frac{SH}{\rho c^2} \sqrt{AK \log(SAH^4 K^{3/2}/\delta)} \right) .$$

*Proof.* Similar to the case with $(s, a)$-rectangular uncertainty set, for any $k$, we can decompose $(\hat{V}_1^{\pi_k} - \hat{V}_1^{\pi_k})(s)$ as,

$$(\hat{V}_1^{\pi_k} - \hat{V}_1^{\pi_k})(s) \le \sum_{h=1}^{H} \mathbb{E}_{\pi_k, \{p_t\}_{t=1}^h} \left[ (r_h^k(s, a) - \hat{r}_h^k(s, a)) + \left( \sigma_{\hat{\mathcal{P}}_h(s)} \left( \hat{V}_{h+1}^{\pi_k} \right) (s, a) - \sigma_{\mathcal{P}_h(s)} \left( \hat{V}_{h+1}^{\pi_k} \right) (s, a) \right) + b_h^k(s, a) \right] .$$

Thus by the design of our bonus function and with probability at least $1 - \delta$, we have

$$\sum_{k=1}^{K} (\hat{V}_1^{\pi_k} - V_1^{\pi_k})(s)$$

$$\le 2 \sum_{k=1}^{K} \sum_{h=1}^{H} \mathbb{E}_{\pi_k, \{p_t\}_{t=1}^h} \left[ b_h^k(s, a) \right]$$

$$= H\sqrt{K} + O \left( \frac{1}{\rho c} \sqrt{S \log(SAH^4 K^{3/2}/\delta)} \right) \sum_{k=1}^{K} \sum_{h=1}^{H} \mathbb{E}_{\pi_k, \{p_t\}_{t=1}^h} \left[ \sqrt{\frac{1}{N_h^k(s, a)}} \right] ,$$

where $c$ is a problem dependent constant.

Let $\omega_h = \{p_t\}_{t=1}^h / P_h^o$, and assume $P_h^o(\cdot \mid s, a) \ge c, \forall(s, a)$. By Lemma E.2, we have the bound of the visitation counts:

$$\sum_{k=1}^{K} \sum_{h=1}^{H} \mathbb{E}_{\pi_k, \{p_t\}_{t=1}^h} \left[ \sqrt{\frac{1}{N_h^k(s, a)}} \right] \le \sum_{k=1}^{K} \sum_{h=1}^{H} \mathbb{E}_{\pi_k, P_h^o} \left[ \omega_h \sqrt{\frac{1}{N_h^k(s, a)}} \right]$$

$$\le \max_{h \in [H]} \omega_h \sum_{k=1}^{K} \sum_{h=1}^{H} \mathbb{E}_{\pi_k, P_h^o} \left[ \sqrt{\frac{1}{N_h^k(s, a)}} \right]$$

$$\le \frac{2H\sqrt{SAK}}{c} .$$

Combining everything, conditioned on the good event we have

$$\sum_{k=1}^{K} (\hat{V}_1^{\pi_k} - V_1^{\pi_k})(s) = O \left( \frac{SH}{\rho c^2} \sqrt{AK \log(SAH^4 K^{3/2}/\delta)} \right) .$$

$\qquad\square$

**Lemma C.5.** *For any $h, k, s, a$, the following inequality holds with probability at least $1 - \delta$,*

$$\sigma_{\hat{\mathcal{P}}_h(s)}(\hat{V}_{h+1}^{\pi_k})(s, a) - \sigma_{\mathcal{P}_h(s)}(\hat{V}_{h+1}^{\pi_k})(s, a) \le \frac{2H}{\rho c} \sqrt{\frac{4S \log(8SAH^4 K^2/\delta'\rho)}{N_h^k(s, a)}} + \frac{1}{\sqrt{K}}.$$

*where $c$ is the minimal element of $P_h^o$.*

*Proof.* By the definition of $\sigma_{\mathcal{P}_h(s)} \left( \hat{V}_{h+1}^{\pi_k} \right)(s, a) = \inf_{P_h \in \mathcal{P}_h} \sum_{s'} P_h(s' \mid s, a) \hat{V}_{h+1}^{\pi_k}(s')$, we consider the following optimization problem:

$$\min_{P_h} \sum_{s'} P_h(s' \mid s, a) \hat{V}_{h+1}^{\pi_k}(s')$$

$$\text{s.t.} \quad \begin{cases} \sum_{s'} P_h(s' \mid s, a) \log \left( \frac{P_h(s'\mid s,a)}{P_h^o(s'\mid s,a)} \right) \le \rho \,, \\ \sum_{s'} P_h(s' \mid s, a) = 1 \,, \\ P_h^o(\cdot \mid s, a) > 0, P_h(\cdot \mid s, a) \ge 0 \,. \end{cases}$$

Let $\tilde{P}_h(s' \mid s, a) = \frac{P_h(s'\mid s,a)}{P_h^o(s'\mid s,a)}$, we can rewrite the above optimization problem as

$$\min_{\tilde{P}_h} \sum_{s'} \tilde{P}_h(s' \mid s, a) P_h^o(s' \mid s, a) \hat{V}_{h+1}^{\pi_k}(s')$$

$$\text{s.t.} \quad \begin{cases} \sum_{s'} \tilde{P}_h(s' \mid s, a') P_h^o(s' \mid s, a') \log \left( \tilde{P}_h(s' \mid s, a) \right) \le \rho \,, \\ \sum_{s'} \tilde{P}_h(s' \mid s, a') P_h^o(s' \mid s, a) = 1 \,, \\ \tilde{P}_h(\cdot \mid s, a) \ge 0 \,. \end{cases}$$

Use the Lagrangian multiplier method and $f(x) = x \log x$, we have the Lagrangian $L(\tilde{P}_h, \eta, \lambda)$ with multiplier $\eta \in \mathbb{R}$, $\lambda \ge 0$,

$$L(\tilde{P}_h, \eta, \lambda)(s, a)$$
$$= \sum_{s'} \tilde{P}_h(s' \mid s, a) P_h^o(s' \mid s, a) \hat{V}_{h+1}^{\pi_k}(s') + \lambda \left( \sum_{s'} \tilde{P}_h(s' \mid s, a') P_h^o(s' \mid s, a') \log(\tilde{P}_h(s' \mid s, a)) - \rho \right)$$
$$- \eta \left( \sum_{s'} \tilde{P}_h(s' \mid s, a) P_h^o(s' \mid s, a) - 1 \right)$$
$$= -\lambda\rho + \eta + \lambda \sum_{s'} P_h^o(s' \mid s, a) \left( f \left( \tilde{P}_h(s' \mid s, a') \right) - \tilde{P}_h(s' \mid s, a') \left( \frac{\eta - V_{h+1}^{\pi_k}(s')}{\lambda} \right) \right).$$

The convex conjugate of $f$ is $f^*(y) = \max_x \langle x, y \rangle - f(x)$. Using $f^*$, we can thus optimize over $\tilde{P}_h$ and rewrite the Lagrangian over as

$$L(\eta, \lambda)(s, a) = \min_{\tilde{P}_h} L(\tilde{P}_h, \eta, \lambda)(s, a) = -\lambda\rho + \eta - \lambda \sum_{s'} P_h^o(s' \mid s, a) f^* \left( \frac{\eta - V_{h+1}^{\pi_k}(s')}{\lambda} \right).$$

Conditioned on $x \ge 0$, $f(x) = x \log x$, notice that the conjugate $f^*(y)$ has the following closed form,

$$f^*(y) = \max_x \langle x, y \rangle - f(x) = \exp(y - 1).$$

Using the closed form of $f^*(y)$, we can rewrite the optimization problem as

$$L(\eta, \lambda)(s, a) = -\lambda\rho + \eta - \lambda \sum_{s'} P_h^o(s' \mid s, a) f^* \left( \frac{\eta - V_{h+1}^{\pi_k}(s')}{\lambda} \right)$$

$$= -\lambda\rho + \eta - \lambda \sum_{s'} P_h^o(s' \mid s, a) \exp \left( \frac{\eta - V_{h+1}^{\pi_k}(s') - \lambda}{\lambda} \right).$$

Taking the derivative of $\eta$,

$$\frac{\partial L}{\partial \eta} = 1 - \sum_{s'} P_h^o(s' \mid s, a) \exp\left(\frac{\eta - V_{h+1}^{\pi_k}(s') - \lambda}{\lambda}\right) = 0,$$

$$\eta = \lambda - \lambda \log\left(\sum_{s'} P_h^o(s' \mid s, a) \exp\left(\frac{-V_{h+1}^{\pi_k}(s')}{\lambda}\right)\right).$$

Directly optimize over $\eta$, we can reduce the problem to

$$L(\lambda)(s, a) = \lambda(1 - \rho) - \lambda \log\left(\sum_{s'} P_h^o(s' \mid s, a) \exp\left(\frac{-V_{h+1}^{\pi_k}(s')}{\lambda}\right)\right) - \lambda,$$

$$= -\lambda\rho - \lambda \log\left(\sum_{s'} P_h^o(s' \mid s, a) \exp\left(\frac{-V_{h+1}^{\pi_k}(s')}{\lambda}\right)\right).$$

Define $g(\lambda, P_h^o) = -L(\lambda)(s, a)$ as

$$g(\lambda, P_h^o) = \lambda\rho + \lambda \log\left(\sum_{s'} P_h^o(s' \mid s, a) \exp\left(\frac{-V_{h+1}^{\pi_k}(s')}{\lambda}\right)\right).$$

Note that the Lagrangian multiplier $\lambda \geq 0$. Then we prove $g$ is bounded within $[-H, H]$ over $[0, H/\rho]$.

$$g(\lambda, P_h^o) = \lambda\rho + \lambda \log\left(\sum_{s'} P_h^o(s' \mid s, a) \exp\left(\frac{-V_{h+1}^{\pi_k}(s')}{\lambda}\right)\right),$$

$$\leq \lambda\rho + \lambda \log\left(\sum_{s'} P_h^o(s' \mid s, a) \exp\left(\frac{-0}{\lambda}\right)\right),$$

$$= \lambda\rho \leq H,$$

where the first inequality follows from $V_{h+1}^{\pi_k}(s') \geq 0$ and the second inequality is by $\lambda \leq H/\rho$.

$$g(\lambda, P_h^o) = \lambda\rho + \lambda \log\left(\sum_{s'} P_h^o(s' \mid s, a) \exp\left(\frac{-V_{h+1}^{\pi_k}(s')}{\lambda}\right)\right),$$

$$\geq \lambda\rho + \lambda \log\left(\sum_{s'} P_h^o(s' \mid s, a) \exp\left(\frac{-H}{\lambda}\right)\right),$$

$$= \lambda\rho - H \geq -H,$$

where the first inequality follows from $V_{h+1}^{\pi_k}(s') \leq H$ and the second inequality is by $\lambda \geq 0$.

Moreover, from the induction above we know that for any $P$, $g(0, P) \leq 0$ and for $\lambda > H/\rho$,

$$g(\lambda, P) \geq \lambda\rho + \lambda \log(\exp(-H/\lambda)) > 0.$$

Therefore, g achieves its minimum over $\lambda \in [0, H/\rho]$. We remark that the same form is also used for sample complexity results ([Badrinath and Kalathil, 2021, Yang et al., 2021]).

We can now rewrite

$$\sigma_{\hat{\mathcal{P}}_h(s)}\left(\hat{V}_{h+1}^{\pi_k}\right)(s, a) - \sigma_{\mathcal{P}_h(s)}\left(\hat{V}_{h+1}^{\pi_k}\right)(s, a) = \min_{0 \leq \lambda_1 \leq H/\rho} g\left(\lambda_1, \hat{P}_h^{o,k}\right) - \min_{0 \leq \lambda_2 \leq H/\rho} g\left(\lambda_2, P_h^o\right)$$

$$\leq \max_{0 \leq \lambda \leq H/\rho} \left| g\left(\lambda, \hat{P}_h^{o,k}\right) - g\left(\lambda, P_h^o\right) \right|.$$

By Nilim and El Ghaoui [2005] (Appendix C), when $\lambda = 0$, $g\left(\lambda, \hat{P}_h^{o,k}\right) = g\left(\lambda, P_h^o\right) = \min_{s \in \mathcal{S}} V_{h+1}^{\pi_k}(s)$. Therefore, it suffice to bound over $\max_{c \le \lambda \le H/\rho} \left| g\left(\lambda, \hat{P}_h^{o,k}\right) - g\left(\lambda, P_h^o\right) \right|$, where $c > 0$. We now have

$$
\left| g\left(\lambda, \hat{P}_h^{o,k}\right) - g\left(\lambda, P_h^o\right) \right|
$$

$$
= \left| \lambda \log \left( \sum_{s'} \hat{P}_h^{o,k}(s' \mid s,a) \exp \left( \frac{-V_{h+1}^{\pi_k}(s')}{\lambda} \right) \right) - \lambda \log \left( \sum_{s'} P_h^o(s' \mid s,a) \exp \left( \frac{-V_{h+1}^{\pi_k}(s')}{\lambda} \right) \right) \right|
$$

$$
= \left| \lambda \log \left( 1 + \frac{\sum_{s'} (\hat{P}_h^{o,k}(s' \mid s,a) - P_h^o(s' \mid s,a)) \exp \left( \frac{-V_{h+1}^{\pi_k}(s')}{\lambda} \right)}{\sum_{s'} P_h^o(s' \mid s,a) \exp \left( \frac{-V_{h+1}^{\pi_k}(s')}{\lambda} \right)} \right) \right|
$$

$$
\le 2\lambda \left| \frac{\sum_{s'} (\hat{P}_h^{o,k}(s' \mid s,a) - P_h^o(s' \mid s,a)) \exp \left( \frac{-V_{h+1}^{\pi_k}(s')}{\lambda} \right)}{\sum_{s'} P_h^o(s' \mid s,a) \exp \left( \frac{-V_{h+1}^{\pi_k}(s')}{\lambda} \right)} \right|
$$

$$
\le 2\lambda \max_{s'} \left| \frac{\hat{P}_h^{o,k}(s' \mid s,a) - P_h^o(s' \mid s,a)}{P_h^o(s' \mid s,a)} \right|
$$

where the first inequality follows from $|\log(1 + x)| \le 2|x|$ and the second inequality follows from the Holder's inequality. By Hoeffding's inequality and an union bound over all $s, a', N_h^k(s,a)$, the following inequality holds with probability at least $1 - \delta$,

$$
\max_{s'} \left| \hat{P}_h^{o,k}(s' \mid s,a) - P_h^o(s' \mid s,a) \right| \le \left\| \hat{P}_h^{o,k}(\cdot \mid s,a) - P_h^o(\cdot \mid s,a) \right\|_1 \le \sqrt{\frac{4S \log(SAH^2 K/\delta)}{N_h^k(s,a)}} .
$$

Then we create an $\epsilon$-net $N_\epsilon(\lambda)$ with $g$ over $\lambda \in [0, H/\rho]$ such that

$$
\max_{\lambda \in [0, H/\rho]} |g(\lambda, \hat{P}_h^{o,k}) - g(\lambda, P_h^o)| \le \max_{\lambda \in N_\epsilon(\eta)} |g(\lambda, \hat{P}_h^{o,k}) - g(\lambda, P_h^o)| + 2\epsilon .
$$

Then we know that $|N_\epsilon(\lambda)|$ is bounded by the area of the rectangle $[0, H/\rho] \times [-H, H]$ over $\epsilon^2$,

$$
|N_\epsilon(\lambda)| \le \frac{2H^2}{\rho \epsilon^2} .
$$

Taking an union bound over $N_\epsilon(\lambda)$ and denote $c = \min_{s'} P_h^o(\cdot \mid s,a)$, we have the following inequality happens with at least $1 - \delta'$ probability:

$$
\begin{aligned}
\sigma_{\hat{\mathcal{P}}_h(s)}(\hat{V}_{h+1}^{\pi_k})(s,a) - \sigma_{\mathcal{P}_h(s)}(\hat{V}_{h+1}^{\pi_k})(s,a) &\le \max_{\lambda \in [0, H/\rho]} |g(\lambda, \hat{P}_h^{o,k}) - g(\lambda, P_h^o)| \\
&\le \max_{\lambda \in N_\epsilon(\lambda)} |g(\lambda, \hat{P}_h^{o,k}) - g(\lambda, P_h^o)| + 2\epsilon \\
&\le 2\frac{H}{\rho} \max_{s'} \left| \frac{\hat{P}_h^{o,k}(s' \mid s,a) - P_h^o(s' \mid s,a)}{P_h^o(s' \mid s,a)} \right| + 2\epsilon \\
&\le 2\frac{H}{\rho c} \sqrt{\frac{4S \log(2SAH^4 K/\delta' \rho \epsilon^2)}{N_h^k(s,a)}} + 2\epsilon ,
\end{aligned}
$$

Take $\epsilon = \frac{1}{2\sqrt{K}}$, then

$$
\sigma_{\hat{\mathcal{P}}_h(s)}(\hat{V}_{h+1}^{\pi_k})(s,a) - \sigma_{\mathcal{P}_h(s)}(\hat{V}_{h+1}^{\pi_k})(s,a) \le 2\frac{H}{\rho c} \sqrt{\frac{4S \log(8SAH^4 K^2/\delta' \rho)}{N_h^k(s,a)}} + \frac{1}{\sqrt{K}} .
$$

$\square$

# D PROOF OF PROPOSITION 1

**Claim D.1** (Suboptimality of non-robust optimal policy). *There exists a robust MDP $\mathcal{M} = \langle \mathcal{S}, \mathcal{A}, \mathcal{P}, r, H \rangle$ with uncertainty set $\mathcal{P}$ of uncertainty radius $\rho$, such that the non-robust optimal policy is $\Omega(1)$-suboptimal to the uniformly random policy.*

*Proof.* We consider a robust MDP with three states $s_0, s_1, s_2$ and two actions $a_0, a_1$. Without loss of generality, we let $s_0$ be the initial state. On the initial state $s_0$, both actions will lead to a reward of $0$. On state $s_1$, a reward of $1/(H-1)$ is given for both actions. On state $s_2$, a reward of $-1/(H-1)$ is given for both actions. The nominal transition dynamic of the MDP is the following. Taking action $a_0$ on $s_0$ will be transited to $s_1$ with a probability of $\epsilon$ and be transited to $s_2$ with a probability of $\epsilon$, while $\epsilon > 0.5$. Taking the other action $a_1$ will have equal probability of transiting to $s_1$ and $s_2$. The states $s_1$ and $s_2$ are absorbing, in the sense that taking any action on these two states will be transited by to the same state. The transition of the MDP is also illustrated in Figure 4, where a dashed line denotes a probabilistic transition and a solid line denotes deterministic transition. With the nominal transition, it is clear that an optimal policy would be always taking $a_0$. Denote

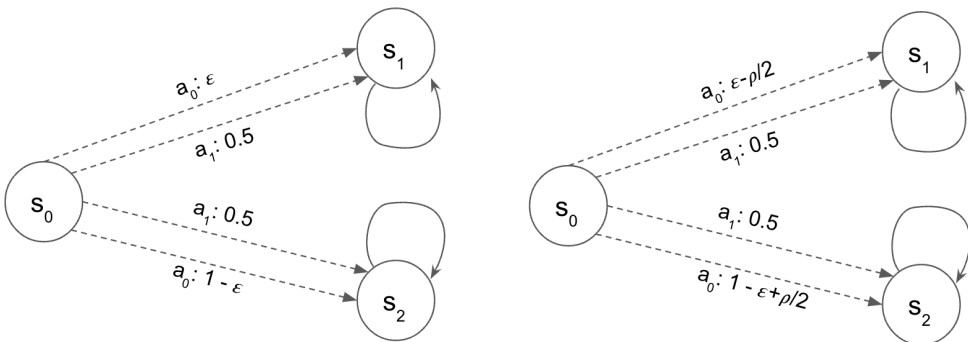

Figure 4: The left figure describes the nominal transition dynamic of the MDP. The right figure describes the robust transition dynamic of the MDP.

this policy as $\pi_{o,*}$, the value for this policy under nominal transition over $K$ episodes is

$$V^{\pi_{o,*}}(s_0) = K(H-1)\left(\epsilon \cdot \frac{1}{H-1} - (1-\epsilon) \cdot \frac{1}{H-1}\right) = 2\epsilon - 1 > 0,$$

where the last inequality is due to $\epsilon > 0$.

However, consider the uncertainty radius $\rho$ and the robust transition denoted by the right figure of Figure 4. That is, taking $a_0$ on $s_0$ will leads to a transition to $s_1$ with probability $\epsilon - \rho/2$ and to $s_2$ with probability $1 - \epsilon + \rho/2$. Note that as $\epsilon > 0.5$, $\rho \leq 1$, $\epsilon - \rho/2 > 0$. Moreover, this transition is indeed the worst case transition for any non-uniform policy. Let $\tilde{V}$ denotes the robust value under the above described transition. With a uniform policy $\pi$, the value of it under this transition is

$$\tilde{V}^{\pi}(s_0) = K(H-1)\left(0.5\left(\epsilon - \frac{\rho}{2}\right) \cdot \frac{1}{H-1} - 0.5\left(1 - \epsilon + \frac{\rho}{2}\right)\right) \cdot \frac{1}{H-1}\right) = \epsilon - \rho/2 - 0.5.$$

The value of $\pi_{o,*}$ is, however,

$$\tilde{V}^{\pi_{o,*}}(s_0) = K(H-1)\left(\left(\epsilon - \frac{\rho}{2}\right) \cdot \frac{1}{H-1} - \left(1 - \epsilon + \frac{\rho}{2}\right)\right) \cdot \frac{1}{H-1}\right) = 2\epsilon - \rho - 1.$$

For any $2\epsilon - 1 \leq \rho \leq 1$, we have $\tilde{V}^{\pi_{o,*}}(s_0) \leq \tilde{V}^{\pi}(s_0)$. Since $\epsilon > 0.5$ is arbitrary, the optimal policy under the nominal transition is non-robust even under the slightest perturbation. $\qquad\square$

# E   AUXILIARY LEMMAS

**Lemma E.1** (Bartlett [2013]). *An $\epsilon$-cover of a subset $T$ of a pseudometric space $(S, d)$ is a set $\hat{T} \subset T$ such that for each $t \in T$ there is a $\hat{t} \in \hat{T}$ such that $d(t, \hat{t}) \leq \epsilon$. The $\epsilon$-covering number of $T$ is*

$$N(\epsilon, T, d) = \min \left\{ |\hat{T}| : \hat{T} \text{ is an } \epsilon\text{-cover of } T \right\} .$$

*Let $F_d$ be the set of L-Lipschitz functions (wrt $\| \cdot \|_\infty$) mapping from $[0,1]^d$ to $[0,1]$. Then*

$$\log N \left( \epsilon, F_d, \| \cdot \|_\infty \right) = \Theta \left( \left( \frac{L}{\epsilon} \right)^d \right) .$$

**Lemma E.2** (Lemma 7.5 Agarwal et al. [2019]). *For arbitrary $K$ sequence of trajectories $\{s_h^k, a_h^k\}_{h=1}^H$, $k = 1, \ldots, K$, we have*

$$\sum_{k=1}^K \sum_{h=1}^H \frac{1}{\sqrt{N_h^k(s_h^k, a_h^k)}} \leq 2H\sqrt{SAK} .$$

*Proof.* We have

$$
\begin{aligned}
\sum_{k=1}^K \sum_{h=1}^H \frac{1}{\sqrt{N_h^k \left( s_h^k, a_h^k \right)}} &= \sum_{h=1}^H \sum_{(s,a) \in \mathcal{S} \times \mathcal{A}} \sum_{i=1}^{N_h^K(s,a)} \frac{1}{\sqrt{i}} \\
&\leq 2 \sum_{h=1}^H \sum_{(s,a) \in \mathcal{S} \times \mathcal{A}} \sqrt{N_h^K(s,a)} \\
&\leq \sum_{h=1}^H \sqrt{SA \sum_{s,a} N_h^K(s,a)} \\
&= H\sqrt{SAK} ,
\end{aligned}
$$

where the first inequality is by $\sum_{i=1}^N \frac{1}{\sqrt{i}} \leq 2\sqrt{N}$ and the second inequality follows by Cauchy-Schwarz inequality. $\quad\square$

**Lemma E.3** (Fundamental inequality of Online Mirror Descent for RL (Lemma 17 Shani et al. [2020])). *Let $\beta > 0$. Let $\pi_h^1(\cdot \mid s)$ be the uniform distribution. Then, by updating with OMD and with KL divergence regularization, for any $k \in [K], h \in [H]$ and $s \in \mathcal{S}$, the following holds for any stationary policy $\pi$,*

$$\sum_{k=1}^K \left\langle Q_h^k(\cdot \mid s), \pi_h^k(\cdot \mid s) - \pi_h(\cdot \mid s) \right\rangle \leq \frac{\log A}{\beta} + \frac{\beta}{2} \sum_{k=1}^K \sum_a \pi_h^k(a \mid s) \left( Q_h^k(s,a) \right)^2 . \tag{7}$$

# F   MORE EXPERIMENTAL DETAILS

**Other configurations and set up**   The episode length is set to 20 and all algorithms are trained with 3000 episodes. The evaluation results are averaged over 20 runs and is presented with 1 standard deviation. All experiments are conducted with 64 core ADM 3990X.

**Results with KL divergence constrained** $(s, a)$**-rectangular uncertainty sets**   With the uncertainty set described with KL divergence, we present the following experimental results. All other configurations and set up remains the same with those for uncertainty set with $\ell_1$ distance.

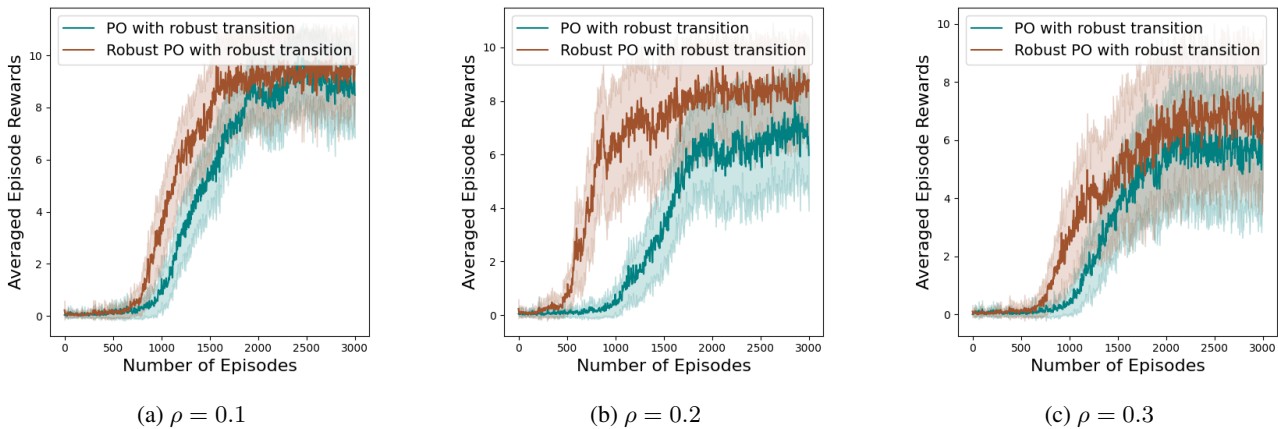

(a) $\rho = 0.1$          (b) $\rho = 0.2$          (c) $\rho = 0.3$

Figure 5: Cumulative rewards obtained by robust and non-robust policy optimization on robust transition with different level of uncertainty $\rho = 0.1, 0.2, 0.3$ under KL divergence.

