# OpenReview forum: "Online Policy Optimization for Robust Markov Decision Process"
_auai.org/UAI/2024/Conference — UAI 2024 poster_

### Official Review · Reviewer_5x3N · 2024-03-08

**Q2-1 Originality-Novelty:** 2
**Q2-2 Correctness-Technical Quality:** 2
**Q2-5 Clarity Of Writing:** 2

**Q1 Summary And Contributions:**

The author study the problem of determining a robust policy in a Markov decision process where the transition function is not certain. They propose an algorithm (Robust Optimistic Policy Optimization) to minimize a cumulative regret measure, and they also give a theoretical bound on this regret. Theoretic results as well as experimental results are given.

**Q2-3 Extent To Which Claims Are Supported By Evidence:**

3: Good: the main claims are supported by convincing evidence (in the form of adequate experimental evaluation, proofs, (pseudo-)code, references, assumptions).

**Q2-4 Reproducibility:**

2: Fair: key resources (e.g. proofs, code, data) are unavailable but key details (e.g. proof sketches, experimental setup) are sufficiently well-described for an expert to confidently reproduce the main results.

**Q3 Main Strengths:**

The topic is interesting and, as far as I know, the contribution is novel. It gives some theoretical and experimental results.

**Q4 Main Weakness:**

I'm not convinced that the contribution is innovative, nor that it will have a significant impact on the field. But maybe it is just not highlighted enough in the paper.

I think this paper needs to be revised and improved, and possibly submitted in another format (see below).

The paper is moderately well written and organized. Many terms or notions are used without being defined, or with a late definition, which makes some parts of the paper harder to understand. Some parts are just a succession of paragraph with no transition in between. Sections 4 and 5 are very technical and do not contain enough explanations. Finally, both sections 4 and 5 are far too dependent on the appendices, which makes even more difficult to read.

The experimental part misses a more detailed discussion part.

I think the authors should consider submitting an extended version of the paper to a journal (as IJAR for example). This would give them the possibility to give more details and explanations, and perhaps (I recommend that) a running example that would make the paper even more understandable.

See below for more details.

**Q5 Detailed Comments To The Authors:**

Please find below some remarks and recommendations, as well as some questions that should, in my opinion, be answered in the paper.

    • The paper cut is, in my opinion, too academic.

    • The authors introduce their algorithm as an optimistic policy optimization. However, they define their optimization criterion as the minimization of a regret measure (the best worst case policy). The regret is a pessimistic criterion, as it assumes that the nature is adversarial and thus always chooses the worst possible dynamic. Maybe I missed something here.

    • Many notations are used without being defined or with a late definition, which makes the understanding of the paper difficult for a non expert of the topic. For example, Δ in page 3, c in definition 3.1, Aρ in definition 3.2, rectangular uncertainty set, the term “tabular”.

    • Is $\mathcal{P}$ continuous or discreet? Is this important in your computations or not?

    • In the sentence “the cumulative regret is defined to be the cumulative difference between the robust value of πk and the robust value of the optimal policy”, what do you mean by optimal? Optimal w.r.t. what criterion?

    • The result of claim 4.1 is not clear as the definition of "Ω(1)-sub-optimal" is not obvious to me.

    • Algorithm 1 does not specify how $\pi_1$ is obtained.

    • I do not understand what the inputs (algorithm 1) correspond to and how they are obtained

    • page 6: "if there is a policy that achieves a high return then this policy achieves a high return." this is a tautology.

    • Concerning the experimental results, I have multiple questions and remarks:

        ◦ I do not understand the relevance of the grid world example here. What is the interpretation on the perturbation of the transition? It seems to me that this only suggest a wrong definition of the transition function.

        ◦ The discussion part is too brief. I would have appreciated more comments. As an example, what is the range of rewards? The difference between PO and Robust PO do not seem to be huge, but this depend on the range. One might ask why we should make more complicated calculations for such a small difference in performance. Another question is: what are we really interested in in this curve? Is it all the reward values from 0 to 3000 episodes? or only at 3000? Is 3000 the usual number of episodes in practice? We can indeed see that after only 1000 episodes, the performance difference is much more significant than after 3000. But if we are only interested in the reward after 3000 episodes, the difference is, in my opinion, not much significant.

    • I do not know if this question is relevant but I'm curious to know why, in such a setting, the authors were not interested in imprecise probabilities? It is not even mentioned.
Minor remarks and typos:

    • page3: the transition kernel P be chosen → can be chosen

    • What is l_1 distance?

    • What is the level of uncertainty? How is it fixed?

    • claim 4.1 is then referred to as proposition 4.1

    • page 4: it motivates us to to propose → remove one to

    • "in addition, to reducing computational complexity" → to reduce

**Q9 Complying With Reviewing Instructions:**

Yes

---

> ### Author Rebuttal · Authors · 2024-04-02
>
> We thank the reviewer for the comments. The minor issues and typographical errors have been corrected in our manuscript.We note that the (??) citation errors are displayed correctly in the supplementary file.  Below, we address the primary concern that has been raised. We are also confused with first comment regarding the paper cut is too academic, as we believe our paper is an academic research paper.
>
> >Q1: The is an optimistic policy optimization, but the optimization criterion is the minimization of a regret measure which is a pessimistic criterion.
>
> We note that the concept of regret is a foundational aspect of online learning, applicable even in stochastic environments. Our study focuses on the finite-horizon Robust MDP with online interactions. We characterize our interactions as 'online' because the learner receives rewards only for the states that are visited, a scenario commonly associated with bandit feedback. This scenario inherently presents the classic exploration-exploitation dilemma, underscoring the necessity for optimistic learning approaches. **The use of 'optimistic' in our title only denotes this learning style and does not imply an adversarial environment.** We direct the reviewer to [1], which provides one of the earliest regret guarantees for online reinforcement learning featuring an optimistic algorithm within a stochastic transition framework.
>
> >Q2: Is $\mathcal{P}$ continuous or discreet?
>
> We define the uncertainty set $\mathcal{P}$ as a set of transition kernels surronding the nominal transition kernel $P$, within a given radius. Consequently, the set is  continuous in general.
>
> >Q3: In the sentence “the cumulative regret is defined to be the cumulative difference between the robust value of $\pi_k$ and the robust value of the optimal policy”, what do you mean by optimal?
>
> We define the optimal value as the optimal cumulative rewards under the worst case transitions
> $$V^{\ast} _h (s) = \max _{\pi} V^{\pi} _h (s) = \max _{\pi} \min _{\{P _h\} \in \{\mathcal{P} _h\}} V^{\pi, \{P _h\}} _h (s)$$
> We note that this is defined in section 3, above the paragraph about Bellman robust equations.
>
> >Q4: The result of claim 4.1 is not clear as the definition of "Ω(1)-sub-optimal" is not obvious to me.
>
> We used the term Ω(1)-sub-optimal to describe that the non-robust policy is **at least a constant factor away from the optimal robust policy**, and thus can be arbitrarily bad (compared to the optimal robust policy).
>
> >Q5: Explain the Details of experiments.
>
> 1) We want to remark that we designed our algorithm under the tabular episodic robust MDP setting, thus the Gridworld environment provides a viable setup to empirically evaluate our algorithms.
>
> 2) As for the rewards setting, we refer the reviewers for section 6 for more details. Once the agent steps on the reward cell, it will receive a reward of 1, and it will receive no rewards otherwise.
>
> We are interested in the curves as it clearly show that our robust algorithm learns much more efficiently for the robust policy (since the brown and green line does not overlap in general). The reward curve is also a common metrics to evaluate practical RL algorithms. We refer to papers like [2] for more examples. The number of episodes can be given, and therefore we believe it is still important to see our algorithm performs better in general, regardless of the number of episodes.
>
> > Q6: I do not know if this question is relevant but I'm curious to know why, in such a setting, the authors were not interested in imprecise probabilities? It is not even mentioned.
>
> We remark that the nominal transition is not revealed to the learner, and thus the learner must estimate the transition (Equation (2)). We therefore do not understand what the reviewer implies by imprecise probabilities.
>
> > Q7: Algorithm 1 does not specify how $\pi_1$ is obtained.
>
> We thank the reviewer for pointing this out, $\pi_1$ is initalized to be an uniform policy (uniform as in taking equal probability for each action).
>
> > Q8: What the inputs (algorithm 1) correspond to and how they are obtained.
>
> The input of the algorithm is the learning rate $\beta$, which is used to update the policy $\pi$, and the bonus function, which is defined in the appendix and used to update $\hat{Q}_h^k$.
>
> > Q9: What is L_1 distance? What is the level of uncertainty? How is it fixed?
>
> The L_1 distance is defined as the sum of the absolute differences between vectors in a space. The level of uncertainty is represented by the uncertainty radius, denoted as $\rho$. This is a problem-dependent constant, determined by the specific conditions of the environment.
>
> **References**
>
> [1] Auer, Peter, Thomas Jaksch, and Ronald Ortner. "Near-optimal regret bounds for reinforcement learning." Advances in neural information processing systems 21 (2008).
>
> [2] Wang, Y., He, H., & Tan, X. (2020, August). Truly proximal policy optimization. In Uncertainty in Artificial Intelligence (pp. 113-122). PMLR.

---

### Official Review · Reviewer_sqGi · 2024-03-23

**Q2-1 Originality-Novelty:** 3
**Q2-2 Correctness-Technical Quality:** 3
**Q2-5 Clarity Of Writing:** 3

**Q1 Summary And Contributions:**

This paper gives a provable policy optimization algorithm designed for robust Markov Decision Processes (MDP) within an online Reinforcement Learning (RL) framework incorporating an uncertainty set. It also presents a finite-sample complexity bound.

**Q2-3 Extent To Which Claims Are Supported By Evidence:**

3: Good: the main claims are supported by convincing evidence (in the form of adequate experimental evaluation, proofs, (pseudo-)code, references, assumptions).

**Q2-4 Reproducibility:**

3: Good: key resources (e.g. proofs, code, data) are available and key details (e.g. proofs, experimental setup) are sufficiently well-described for competent researchers to confidently reproduce the main results.

**Q3 Main Strengths:**

The paper offers a non-asymptotic regret bound and exhibits sub-linear sample complexity, enhancing the field's understanding. Moreover, the presented sample complexity also reinstates a non-robust policy optimization finding from previous research when the uncertainty set is involved. The paper substantiates its claims through conducted experiments, validating the results obtained.

**Q4 Main Weakness:**

There seem to missing references in the main paper - I see a lot ?? so it makes it a bit harder to follow what is being talked about.

**Q5 Detailed Comments To The Authors:**

See Q4

**Q9 Complying With Reviewing Instructions:**

Yes

---

> ### Author Rebuttal · Authors · 2024-04-02
>
> We thank the reviewer for their insightful comments and constructive suggestions. We apologize for the typographical errors in the citations, and they will be corrected in our revised manuscript. We note that the (??) citation errors are displayed correctly in the supplementary file.

---

### Official Review · Reviewer_Ye9e · 2024-03-25

**Q2-1 Originality-Novelty:** 2
**Q2-2 Correctness-Technical Quality:** 3
**Q2-5 Clarity Of Writing:** 3

**Q1 Summary And Contributions:**

This paper considers the robust MDP problem. Previous works on robust MDP assume access to a generative model and ignore the exploration issue. To address this issue, this paper considers the online setting where the learner can roll out policies in the environment. To solve this online robust MDP problem, this paper proposes a robust optimistic policy optimization algorithm that is provably efficient and provides the first regret bound. In particular, the proposed algorithm incorporates a new bonus term derived by Fenchel conjugates.

**Q2-3 Extent To Which Claims Are Supported By Evidence:**

3: Good: the main claims are supported by convincing evidence (in the form of adequate experimental evaluation, proofs, (pseudo-)code, references, assumptions).

**Q2-4 Reproducibility:**

2: Fair: key resources (e.g. proofs, code, data) are unavailable but key details (e.g. proof sketches, experimental setup) are sufficiently well-described for an expert to confidently reproduce the main results.

**Q3 Main Strengths:**

1. This paper provides the first provably efficient algorithm for solving the online robust MDP problem.
2. This paper is overall well-written and provides a clear summary of its main contributions. It is easy for me to follow this paper.

**Q4 Main Weakness:**

1. Compared with previous theoretical works on robust MDP, this paper poses an additional assumption $P^{o}_h (\cdot|s, a) \geq c > 0$. This assumption requires that any state can be visited by starting from an arbitrary state-action pair, which is not practical. Under this realizability assumption, we may not need to perform strategy exploration, which weakens the main contribution on exploring in the robust MDP.  Besides, the achieved regret bound depends on $c$. As a result, the regret bound achieved in the robust MDP cannot recover that in the non-robust case.
2. This paper does not provide a lower bound for the online robust MDP problem. Therefore, it is unclear whether the attained regret bound is tight or not. Compared with the regret bound in the non-robust case, the attained regret bound is loose in the dependence of $S$.

**Q5 Detailed Comments To The Authors:**

1. The references in this paper do not show normally.

**Q9 Complying With Reviewing Instructions:**

Yes

---

> ### Author Rebuttal · Authors · 2024-04-02
>
> We thank the reviewer for their comments and suggestions. Below, we address the primary concerns that have been raised. The minor issues and typographical errors have been corrected in our manuscript. We also want to note that the (??) citation errors are displayed correctly in the supplementary file.
>
> >Q1: This paper poses an additional assumption $P_h^o(\cdot \mid s,a) > 0$. It requires that any state can be visited by starting from an arbitrary state-action pair.
>
> **A1:** We want to note that this assumption has also been implicitly used in previous analyses of robust MDPs, such as in [1]. For MDPs that do not satisfying this assumption, they  can be slightly perturbed to meet the assumption. Specifically, one can add an arbitrary noise of $\epsilon/(SAH)$ to each transition probability $P_h^o(s' \mid s,a)$. If the rewards are bounded within $[0,1]$, then the robust value function will incur a final error bound of $\epsilon$.
>
> >Q2: This paper does not provide a lower bound for the online robust MDP problem.
>
> **A2:** Although establishing a lower bound for the online robust MDP remains an open challenge, we want to remark on the difficulty and the necessity of designing robust algorithms for online robust MDPs through claim 4.1 in our manuscript. Our claim 4.1 illustrates the importance of adopting a robust policy to minimize regret in robust MDPs (while interacting with the nominal transition). Specifically, it posits that **the optimal policy under the nominal environment could perform even worse than a uniformly random policy.** We highlight the potential pitfalls of neglecting robustness in policy design in this claim.
>
> **References**
>
> [1] Yang W, Zhang L, Zhang Z. Towards theoretical understandings of robust markov decision processes: Sample complexity and asymptotics[J]. arXiv preprint arXiv:2105.03863, 2021.

---

### Official Review · Reviewer_ZjKr · 2024-03-26

**Q2-1 Originality-Novelty:** 2
**Q2-2 Correctness-Technical Quality:** 2
**Q2-5 Clarity Of Writing:** 3

**Q1 Summary And Contributions:**

This paper highlights that despite the success of Reinforcement learning (RL), deploying RL models in real-world scenarios is challenging due to their sensitivity to environmental changes. The robust Markov decision process (MDP) framework addresses this by considering uncertainty around a nominal model. In contrast to existing approaches which need to assume access to the generative model,  this work focuses on efficient online robust MDPs. The proposed robust optimistic policy optimization algorithm incorporates Fenchel conjugates to handle adversarial uncertainty. This study establishes that regret is bound for robust online MDPs.

**Q2-3 Extent To Which Claims Are Supported By Evidence:**

2: Fair: the main claims are somewhat supported by evidence (but the experimental evaluation may be weak, or does not match entirely with the claims, important baselines may be missing, proofs contain important ideas but lack rigor, algorithmic details are only discussed superficially, references are imprecise, assumptions are not sufficiently motivated or explicated, etc.).

**Q2-4 Reproducibility:**

2: Fair: key resources (e.g. proofs, code, data) are unavailable but key details (e.g. proof sketches, experimental setup) are sufficiently well-described for an expert to confidently reproduce the main results.

**Q3 Main Strengths:**

- The considered problem is relevant in practice.
- The paper is easy to read.

**Q4 Main Weakness:**

- The core contribution is not clear, is it about developing an online algorithm or the problem formulation in the context of robust MDP?
- How to design the rectangular sets used to define the uncertainty set?
- The proposed approach is limited to tabular settings.
- If the transition probabilities lie in a set, which one to pick to define the optimal value function V* ?
- What are the exact technical challenges in doing the technical analysis of this work, please mention it clearly.
- Will V* will also be a function of T? in the online setting, it should also change with time ?

**Q5 Detailed Comments To The Authors:**

please check the weakness above.

**Q9 Complying With Reviewing Instructions:**

Yes

---

> ### Author Rebuttal · Authors · 2024-04-02
>
> We thank the reviewer for their insightful comments. In the following, we address the main concern raised.
>
> >Q1: The core contribution of this paper is not clear, is it about developing an online algorithm or the problem formulation in the context of robust MDP?
>
> **A1:** We thank the reviewer for the suggestion and will revise our manuscript to highlight the core contributions of the paper more clearly. We wish to emphasize that **our primary contribution lies in the development of the first algorithm and the associated regret bound for online robust MDPs**. The formulation of robust MDP are not our contribution: it have been extensively investigated separately in prior works.
>
> >Q2: How to design the rectangular sets used to define the uncertainty set?
>
> **A2:** In practical application, both the radius and the choice of uncertainty set (L1/KL, s,a-rectangular/s-rectangular) are a part of the tunable parameters. This flexibility **allows for the adjustment of these parameters** based on the specific problem at hand, aiming for optimal empirical results and enhanced robustness. The concept of a rectangular uncertainty set was initially introduced by [1], drawing inspiration from the structure of recursive multiple priors [2]. Additionally, we wish to highlight that, generally, **robust MDPs become intractable without certain assumptions**, such as rectangularity.
>
> >Q3: The proposed approach is limited to tabular settings.
>
> **A3:**  We would like to remark that our work is the first to address online robust MDPs, and therefore, we chose to start with the most fundamental tabular setting. However, we believe our algorithm has the potential to be extended to settings with function approximation. For example, for problems with a linearly structured transition kernel, we could potentially extend our approach of policy optimization and rederive the inner minimization for the policy evaluation step using the same dual reformulation technique.
>
> >Q4: If the transition probabilities lie in a set, which one to pick to define the optimal value function $V^*$?
>
> **A4:** The optimal value function is defined as $V^{\ast} _h (s) = \max _{\pi} V^{\pi} _h (s) = \max _{\pi} \min _{\{P _h\} \in \{\mathcal{P} _h\}} V^{\pi, \{P _h\}}  _h (s)$, which is the maximum cumulative rewards possible in the case of worst case transitions. We note that **this is defined in section 3 in our paper** (right above the section of robust Bellman equation).
>
> >Q5: What are the exact technical challenges in doing the technical analysis of this work?
>
> **A5:** To summarize, the challenges of our analysis lie in two aspects:
>
> (1) **Common value difference lemma in non-robust policy optimization cannot be applied**. Unlike non-robust policy optimization, the analysis of robust policy optimization is highly dependent on varying robust transitions. The common value difference lemma (Lemma 1 of [4], Lemma 4.2 of [5]) can no longer be directly applied. Naively employing a recursive relation with respect to a fixed transition kernel may lead to linear regret. Thus, **we instead perform recursion conditioned on varying transition kernels**. In this scenario, maintaining optimism is challenging as the expectation at each time step $h$ is taken with respect to a different transition kernel.
>
> (2) **We need to address multiple uncertainties simultaneously in robust MDP setup**. Unlike non-robust policy optimization, the learner now has to tackle the uncertainty from both limited interactions and the robust MDP simultaneously. To establish an optimism bonus for the uncertainties of the transition caused by limited interaction and the uncertainty set, we **derive the dual formulation of the inner optimization problem** $\sigma _{\hat{\mathcal{P}} _{(s,a)}}(V)$. This approach allows us to **decouple the uncertainties and bound each source of uncertainty separately**. We then demonstrate that the dual variable $\eta$ must be bounded at its optimum by examining certain pivot points and by leveraging the convexity of the dual.
>
> >Q6: Will $V^*$ will also be a function of $T$? in the online setting, it should also change with time?
>
> **A6:**  Note that we use $H$ to denote the length of the horizon in each episode, and $K$ to represent the number of episodes (i.e., the number of iterations for the algorithms). The optimal value function, $V^*$, depends on $H$, as it is the optimal cumulative rewards over $H$ steps under worst case transitions.
>
> **References**
>
> [1] Iyengar, G. CORC Tech Report TR-2002-07 Robust dynamic programming.
>
> [2] Epstein, L. G., & Schneider, M. (2003). Recursive multiple-priors. Journal of Economic Theory, 113(1), 1-31.

---

### Official Review · Reviewer_4czo · 2024-03-27

**Q2-1 Originality-Novelty:** 3
**Q2-2 Correctness-Technical Quality:** 3
**Q2-5 Clarity Of Writing:** 3

**Q1 Summary And Contributions:**

This paper presents an online policy optimization algorithm for robust MDPs. The idea is that while one is learning online (i.e., trying to minimize regret), one would like to be robust w.r.t. transition model uncertainty. While previous work has done sample complexity analysis (with access to a generative model), it seems this is the first paper on online robust RL. The paper introduces a notion of `robustified' regret which is quite natural. Bellman updates in standard Q-iteration are replaced by robust Bellman updates (which seems natural enough). The main detail to figure out is really the bonus term. An order expression for it is given possibly because the algorithm and the results are not very sensitive to it. Theoretical results then provide regret bounds with different models of uncertainty. Empirical results on toy problems are presented to validate the gain of robustification.

**Q2-3 Extent To Which Claims Are Supported By Evidence:**

3: Good: the main claims are supported by convincing evidence (in the form of adequate experimental evaluation, proofs, (pseudo-)code, references, assumptions).

**Q2-4 Reproducibility:**

3: Good: key resources (e.g. proofs, code, data) are available and key details (e.g. proofs, experimental setup) are sufficiently well-described for competent researchers to confidently reproduce the main results.

**Q3 Main Strengths:**

+ The ideas in this paper, while somewhat natural, are nice and novel. It seems online learning for robust MDPs has not been done before.

+ The paper is written well except there are many missing (??) equation references. At times, this did make it a bit hard to follow the argument.

**Q4 Main Weakness:**

- (3) and (4) are a bit hard to follow, perhaps the authors can explain things a bit to ease the burden on the reader.

- I am not convinced this would work beyond toy problems as robustification puts considerable computational burden.

**Q5 Detailed Comments To The Authors:**

1. Please fix the many many typos (??)
2. Please explain (3) and (4).
3. Please explain how the uncertainty sets are constructed.

**Q9 Complying With Reviewing Instructions:**

Yes

---

> ### Author Rebuttal · Authors · 2024-04-02
>
> We thank the reviewers' comments and valuable suggestions. We apologize for the typographical errors in the citations, and they will be corrected in our revised manuscript. We note that the (??) citation errors are displayed correctly in the supplementary file. We address the reviewers' questions in more detail as follows:
>
> >Q1: Explain the formula (3) and (4).
>
> **A1:** We appreciate the reviewer's feedback regarding the need for clearer explanation of the formula (3) and (4). Equations (3) and (4) **constitute the dual formulation** for computing $\hat{Q}_h^k$ in the algorithm. To compute $\hat{Q}_h^k$, one must solve a constrained optimization problem, with the constraint linked to the uncertainty set. Instead of directly solving this, where the constraint dimension is $|\mathcal{S}|\times |\mathcal{A}| \times |\mathcal{S}|$, we propose solving it by addressing the dual formulation (Equations (3) and (4)). This approach allows for efficient solutions **using bisection or sub-gradient methods**, significantly reducing the problem's dimension. Due to space constraints, the detailed derivation of Equations (3) and (4) is presented not in the main paper but in the Appendix (Lemmas A.4 and B.3). The high-level idea for their derivation begins with obtaining the primal-dual form **through the Lagrangian multiplier method**, followed by optimizing out the primal variable. For example, in the $(s,a)$-rectangular case, we express the Lagrangian form of $\sigma _{\mathcal{P} _{s,a}} (\hat{V} _{h+1}^{\pi_k})(s,a)$ as
>
> $$\sum_{s^\prime} \tilde{P} _h(s^\prime \mid s,a) P _h^o(s^\prime \mid s,a) \hat{V} _{h+1}^{\pi_k}(s^\prime)  + \lambda \left( \sum _{s^\prime} | \tilde{P} _h(s^\prime \mid s,a)  - 1| P _h^o(s^\prime \mid s,a)  - \rho \right) - \eta \left(\sum _{s^\prime} \tilde{P} _h(s^\prime \mid s,a) P _h^o(s^\prime \mid s,a) - 1 \right),$$
> where $\eta, \lambda$ are both Lagrangian multipliers. Under the characterization of $\ell_1$ distance, we can use the convex conjugate of $f(x) = |x - 1|$ to optimize out $\tilde{P}$, resulting in Equation (3).
>
> >Q2: This method would work beyond toy problems as robustification puts considerable computational burden.
>
> **A2:** We note that the algorithm can be implemented efficiently, as Equations (3) and (4) can be solved effectively **using off-the-shelf optimization methods**, such as bisection or sub-gradient methods. As our work represents the first exploration of robust online MDP, we believe our results constitute an initial step towards the development of efficient and scalable algorithms. We leave the development of more efficient methods as a direction for future research.
>
> >Q3: Explain how the uncertainty sets are constructed.
>
> **A3:** In practical applications, both the radius and the choice of uncertainty set (L1/KL, s,a-rectangular/s-rectangular) are tunable parameters. Practitioners can **select and adjust these parameters** based on the specific problem to achieve better empirical results and enhanced robustness. The theoretical concept of the rectangular uncertainty set was first introduced by [1], inspired by the structure of the recursive multiple prior model [2]. It is also important to note that robust MDPs **generally become intractable without certain assumptions**, such as rectangularity.
>
> **References**
>
> [1] Iyengar, G. CORC Tech Report TR-2002-07 Robust dynamic programming.
>
> [2] Epstein, L. G., & Schneider, M. (2003). Recursive multiple-priors. Journal of Economic Theory, 113(1), 1-31.

---

### Meta-Review · Area_Chair_cZsu · 2024-04-13

This paper investigates the problem of robust MDP, in which the transition probabilities belong to an uncertainty set around a nominal model.  The authors introduce new algorithms by tackling several technical difficulties. The reviewers unanimously agree that this work is of high quality. Congratulations to the authors for this nice work! The authors are also encouraged to take reviewers' feedback into account to enhance the clarity of the paper and improve discussions of prior works further.